# Glycerol enhances mitochondrial metabolism and inflammatory response in pro-inflammatory macrophages

Manami Tanaka[1], Takako Hishiki[2,3], Tomomi Matsuura[3], Masato Yasui[1], Shunsuke Chikuma[4] & Mariko Hara-Chikuma [1✉]

## Abstract

**Although glycerol is a ubiquitous metabolite in mammalian systems, its cellular metabolic pathways and functions have not been fully elucidated. Here, we find that elevated extracellular glycerol modulates intracellular metabolism and pro-inflammatory responses of macrophages. In pro-inflammatory macrophages stimulated with lipopolysaccharide, glycerol is taken up through glycerol channels including Aquaporin 3 (AQP3) and metabolized to glycerol-3-phosphate (G3P), which is then converted to dihydroxyacetone phosphate by glycerol-3-phosphate dehydrogenase 2 (GPD2). This glycerol-driven pathway enhances mitochondrial ATP production, potentially by supplying electrons to the electron transport chain (ETC) via GPD2, and by upregulating the transcription of genes encoding ETC complexes.  In addition, glycerol supplementation elevates intracellular acetyl-CoA levels, promotes histone acetylation at the promoters of pro-inflammatory cytokine genes, and consequently increases cytokine gene expression, suggesting enhanced pro-inflammatory response. In vivo experiments, macrophage-specific AQP3 conditional knockout mice exhibit reduced weight gain and adipose tissue inflammation in a high-fat diet-induced obesity model. Our findings provide novel insights into the metabolic regulation and macrophage inflammation by extracellular glycerol.**

**Keywords** Glycerol; Macrophage; Inflammation; Metabolism; Obesity
**Subject Categories** Immunology; Membranes & Trafficking; Metabolism

## Introduction

Glycerol is primarily derived from the lipolysis of triglycerides, which represent the major form of fat storage and dietary intake in mammals (Brisson et al, 2001; Robergs and Griffin, 1998; Xue et al, 2017). Although glycerol can be synthesized intracellularly from glycerol-3-phosphate (G3P) via glycerol-3-phosphate phosphatase (G3PP) (Mugabo et al, 2016), this pathway contributes minimally to glycerol production, with lipolysis serving as the predominant source. Indeed, circulating glycerol concentrations are low under healthy physiological conditions (<0.1 mmol/L) but increase under lipolytic states such as prolonged exercise and obesity, reaching ~0.3 mmol/L in humans (Robergs and Griffin, 1998). In mice, circulating glycerol is typically ~0.2–0.3 mmol/L in lean controls and increases in diet-induced obesity (e.g., ~0.3–0.4 mM), depending on strain and sampling conditions (Iena et al, 2020). Importantly, while circulating glycerol remains in the sub-millimolar range, local interstitial levels in adipose and muscle tissues can reach several millimolar (2–4 mM), about 30-fold higher than in plasma (Maggs et al, 1995), where immune cells may encounter higher glycerol availability. Despite being ubiquitously present and readily available as a potential energy substrate, the metabolic utilization and functional significance of glycerol in immune cells, including macrophages, remain poorly understood (Xue et al, 2017).

Macrophages are central regulators of innate immunity and inflammation, and play vital roles in pathogen defense, tissue homeostasis, and organ functions (Cox et al, 2021; Mosser et al, 2021). Pro-inflammatory macrophages, historically referred to as the M1-type macrophages, are essential for initiating and sustaining inflammatory responses, and this effector transition is tightly coupled to profound metabolic reprogramming  (Murray et al, 2014; Yunna et al, 2020). Classical activation by lipopolysaccharide (LPS) drives a shift from oxidative phosphorylation (OXPHOS) to glycolysis, along with rewiring of the tricarboxylic acid (TCA) cycle and increased acetyl-CoA–dependent histone acetylation, which promotes inflammatory gene expression (Langston et al, 2019; Lauterbach et al, 2019). Mitochondrial metabolism, particularly through the electron transport chain (ETC), has emerged as a critical determinant of macrophage polarization, ROS production, and inflammatory signaling (Kumar et al, 2024; Wculek et al, 2023; Yin and O'Neill, 2021). These metabolic transitions have been increasingly recognized as important drivers of macrophage function and disease pathogenesis (Appari et al, 2018; Castoldi et al, 2015; Shapouri-Moghaddam et al, 2018; Wculek et al, 2022).

[1]Department of Pharmacology, School of Medicine, Keio University, Tokyo, Japan. [2]Department of Biochemistry, School of Medicine, Keio University, Tokyo, Japan. [3]Clinical Translational Research Center, Keio University Hospital, Tokyo, Japan. [4]Institute of Biotechnology, College of Life Sciences and Medicine, National Tsing-Hua University, Hsinchu, Taiwan. ✉E-mail: mariko.chikuma@keio.jp

In contrast, recent work has challenged the universality of this classical paradigm. Ran et al. showed that genetic deletion of the mitochondrial pyruvate carrier does not impair LPS-induced inflammatory activation, despite markedly reducing glucose flux into mitochondria (Ran et al, 2023). Ball et al. demonstrated that OXPHOS inhibition is not inherently required for macrophage activation, and that pro-inflammatory responses can occur across diverse mitochondrial respiratory states depending on the stimulus and context (Ball et al, 2025). These findings indicate that macrophage activation is compatible with multiple metabolic phenotypes, and that mitochondrial signals supporting inflammatory response can emerge without a stereotypical collapse of OXPHOS.

Glycerol transport across the cell membrane is facilitated by specific aquaporin channels, notably aquaporin 3 (AQP3), AQP7, and AQP9, which function as selective glycerol channels. Macrophages express AQP3 and AQP9, with AQP3 being particularly well characterized for its ability to transport not only glycerol but also water and hydrogen peroxide (Hara-Chikuma et al, 2012; Miller et al, 2010). Although AQP3-mediated glycerol transport has been studied in various cell types, its specific role in cellular processes, including in macrophages, remains largely unexplored (Hara and Verkman, 2003). Our previous study suggested that glycerol transported by AQP3 contributes to ATP production in keratinocytes; however, the underlying metabolic pathways are yet to be elucidated (Hara-Chikuma and Verkman, 2008).

In this study, we aimed to clarify the functional role of elevated extracellular glycerol—specifically reflecting the levels found in lipolytic adipose tissue microenvironments—on the cellular metabolism and inflammatory response of pro-inflammatory macrophages. We found that under high concentrations of extracellular glycerol, glycerol uptake through AQP3 promoted mitochondrial ATP production, accompanied by the upregulation of ETC genes via the G3P shuttle (GPS). We also demonstrated that elevated extracellular glycerol affects inflammatory responses in pro-inflammatory macrophages. In vivo experiment, macrophage-specific AQP3 conditional knockout mice exhibited reduced weight gain and adipose tissue inflammation in a high-fat diet-induced obesity model. These findings uncover a previously unrecognized immunometabolic function of extracellular glycerol in regulating mitochondrial activity and inflammatory responses and highlight AQP3 as a key regulator of inflammatory macrophage activation and systemic metabolic disturbances.

## Results

### Glycerol uptake and its metabolic pathway in pro-inflammatory macrophages

Although the canonical pathways of glycerol metabolism have been characterized in mammalian cells, how glycerol is utilized and metabolically processed in macrophages remains poorly understood. Bone marrow cells from C57BL/6 mice were differentiated into macrophages (BMDMs) in macrophage colony-stimulating factor (M-CSF)-containing medium supplemented with 10% fetal bovine serum (FBS) for 1 week. LPS stimulation for 24 h robustly polarized these BMDMs (M0 macrophages) into pro-inflammatory macrophages with marked induction of tumor necrosis factor-alpha

(TNF-α), interleukin-6 (IL-6), and interleukin-1 beta (IL-1β) (Murray et al, 2014; Orecchioni et al, 2019) (Fig. EV1A). To examine how glycerol affects macrophages, we first confirmed that the FBS used in this study contained ~842 μM glycerol (Fig. EV1B). LPS-primed pro-inflammatory macrophages showed no appreciable alterations in inflammatory marker expression or metabolic activity after 24 h of incubation in either FBS-free or dialyzed FBS (dFBS)-containing medium, whereas the LPS responsiveness of M0 macrophages was affected by the presence or absence of FBS (Figs. EV1C,D, EV2G). Therefore, cells were incubated in FBS-free medium for at least 1 h prior to the assay, a condition in which extracellular glycerol was below the detection limit (<1 μM) (Fig. EV1B), ensuring effective glycerol deprivation in this study (Appendix Table S1).

LPS-primed pro-inflammatory macrophages were incubated with $^{14}$C-labeled glycerol for 3 min to assess the initial rate of glycerol uptake, as previously described (Yang and Verkman, 1997). The uptake of $^{14}$C glycerol was significantly increased at concentrations above 100 μM but not at 10 μM, compared to 1 μM. This result suggests that cellular glycerol concentrations in pro-inflammatory macrophages may range between 10 and 100 μM (Fig. 1A). When macrophages were incubated with 1 mM glycerol, treatment with either an AQP3 inhibitor DFP00173 (DFP) (Sonntag et al, 2019) or an AQP3-blocking monoclonal antibody (Hara-Chikuma et al, 2020) significantly reduced intracellular glycerol levels, indicating that extracellular glycerol is partially transported by AQP3 (Fig. 1B).

We next investigated the intracellular metabolic fate of glycerol using stable isotope tracing and metabolomics (De Jesus et al, 2022; Tomas-Gamisans et al, 2019). LPS-primed pro-inflammatory macrophages were treated with $^{13}$C$_3$-labeled glycerol for 3 h, followed by extraction and metabolite analysis. $^{13}$C-labeled metabolites detected in macrophages are shown in Fig. 1C. As shown in Fig. 1D, $^{13}$C incorporation was most prominent in G3P, followed by dihydroxyacetone phosphate (DHAP), fructose 1,6-bisphosphate (FBP), 3-phosphoglycerate (3PG), and acetyl-CoA. Glycerol supplementation significantly increased labeled G3P, DHAP, FBP, and 3PG, reflecting its entry into the G3P shuttle and partial incorporation into glycolytic intermediates. Importantly, glycerol supplementation also elevated acetyl-CoA levels, suggesting a potential enhancement of mitochondrial flux and substrate availability for histone acetylation. Glucose uptake, as assessed by both [$^3$H]-glucose and fluorescent-labeled glucose derivative, was not affected by glycerol supplementation (Fig. EV2A,B). We also verified that glycerol uptake was independent of the presence or absence of glucose (Fig. EV2C).

### Glycerol induces mitochondrial ATP production via GPS

The intracellular metabolic state of LPS-stimulated macrophages has been reported to be highly dependent on the experimental context and duration of stimulation. Although previous studies described that LPS-stimulated BMDMs undergo profound metabolic reprogramming, characterized by reduced mitochondrial oxidative phosphorylation (OXPHOS) and increased glycolysis (Langston et al, 2019; Lauterbach et al, 2019), more recent reports have demonstrated divergent effects on mitochondrial respiration (Ball et al, 2025; Ran et al, 2023). Because supplementation with high concentrations of glycerol altered intracellular metabolic

intermediates, we quantified ATP production from mitochondrial respiration and glycolysis by measuring the oxygen consumption rate (OCR) and extracellular acidification rate (ECAR) using the Seahorse XF Real-Time ATP Rate Assay. We confirmed under our experimental conditions that LPS-primed pro-inflammatory macrophages underwent a metabolic shift toward glycolysis, resulting in a marked reduction of mitochondrial ATP production (Fig. EV2F). We set extracellular glycerol to 1 mM to approximate a glycerol-rich lipolytic tissue microenvironment that can exceed circulating levels, and to assess glycerol-driven metabolic remodeling and transcriptional changes in pro-inflammatory macrophages. After 3-h incubation with high concentrations of glycerol in pro-inflammatory macrophages, both mitochondrial and glycolytic-ATP production were increased, with a greater enhancement observed in mitochondrial ATP production (Fig. 2A,B). After 24 h of incubation, only mitochondrial ATP production remained elevated (Fig. 2C). These results indicate that high concentrations of glycerol modulated the metabolic state of LPS-primed macrophages, leading to an increase in mitochondrial ATP production despite the LPS-induced reduction in OXPHOS.

As shown in Fig. 1D, glycerol markedly increased intracellular levels of G3P. G3P can be metabolized to glycerolipids (Xue et al, 2017), dephosphorylated at G3PP to produce glycerol (Mugabo et al, 2016), or utilized in the G3P shuttle (GPS) (Possik et al, 2021). In the GPS pathway, G3P is oxidized by GPD2, a rate-limiting enzyme in the outer mitochondrial membrane, to generate DHAP, which provides electrons to the electron transport chain (ETC) and promotes ATP production (Eriksson et al, 1995; Liu et al, 2021; Oh et al, 2023; Oh et al, 2024). To test the hypothesis that glycerol affects GPS and leads to increased mitochondrial ATP production, three key steps were targeted for inhibition: glycerol uptake by AQP3, the phosphorylation of glycerol to G3P by glycerol kinase (GK), and the oxidation of G3P to DHAP by GPD2 (Fig. 2D). The inhibitors used were DFP for AQP3, 1-thioglycerol (thio) for GK (Zhou et al, 2023), and KM04416 (KM) for GPD2 (Oh et al, 2023). When cells were exposed to a high extracellular glycerol concentration (1 mM), these inhibitors suppressed the glycerol-induced increases in OCR and mitochondrial ATP production, suggesting the involvement of glycerol-induced GPS in mitochondrial ATP production (Fig. 2E–G). In contrast, under low extracellular glycerol levels (serum-starved control conditions), the inhibitors did not affect OCR and ATP production rate (Fig. 2E–G). In addition, glycerol at concentrations below 300 μM did not affect either mitochondrial or glycolytic-ATP production in pro-inflammatory macrophages (Fig. EV2H). These results indicate that under high-glycerol conditions, glycerol is actively transported through AQP3, metabolized via the GPS pathway, and utilized to support mitochondrial ATP production.

To further confirm the role of glycerol-mediated G3P in cellular metabolism, we overexpressed GK in inflammatory macrophages via adenoviral gene transfer (Fig. EV2D). This intervention led to increased OCR and mitochondrial ATP production in the presence of glycerol (Fig. EV2E), supporting the conclusion that glycerol metabolism through GK and GPD2 enhances mitochondrial function.

## Glycerol affects gene expressions related to mitochondrial respiration *via* the GPS

Next, we determined the effect of glycerol on gene expression, potentially influencing cellular metabolism. LPS-primed pro-

inflammatory macrophages were treated with glycerol (1 mM) for 6 h following glycerol starvation, and RNA sequencing (RNA-seq) was performed. Principal component analysis (PCA) of the RNA sequencing data revealed a distinct clustering of glycerol-treated pro-inflammatory macrophages compared to control cells (Fig. 3A). Differentially expressed gene (DEG) analysis confirmed transcriptional changes induced by glycerol treatment. Among 18,438 genes with valid Entrez Gene IDs, DESeq2 identified 385 upregulated and 411 downregulated genes using a Benjamini–Hochberg adjusted $p$ value ($p$adj) <0.05 as the significance threshold (Appendix Tables S2, S3). KEGG pathway analysis of these DEGs revealed that glycerol supplementation predominantly upregulated genes involved in mitochondrial respiration (Fig. EV3A) (Billingham et al, 2022; Mills et al, 2017; Urra et al, 2017). Heatmap analysis of mitochondrial respiration–related genes showed that 6-h glycerol exposure upregulated ETC Complexes I, II, and III, which are key components of the inner mitochondrial membrane involved in electron transport during OXPHOS (Fig. 3B; Appendix Table S4) (Lunt and Vander Heiden, 2011). Notably, quantitative PCR analysis verified that upregulation of ETC-related genes was already detectable at 3 h after glycerol supplementation, including *Ndufa2*, *Ndufa6*, and *Ndufb9* (Complex I); *Sdha* and *Sdhb* (Complex II); and *Uqcc2*, *Uqcc3*, and *Uqcrc1* (Complex III) (Fig. 3C–E). By contrast, these glycerol-induced changes in ETC-related gene expression were not observed in M0 macrophages, even under high-glycerol conditions (Fig. EV3B). Furthermore, we observed that inhibiting the breakdown of glycerol to G3P and DHAP suppressed glycerol-induced gene expressions related to Complex III, indicating a role for GPS in regulating glycerol-induced ETC Complex expression (Fig. 3F). These findings suggest that glycerol alters the expression of the ETC complex-related genes in pro-inflammatory macrophages via GPS, potentially influencing intracellular metabolism.

## Extracellular glycerol promotes histone acetylation for inflammatory cytokines

In M0 macrophages, LPS stimulation increases intracellular acetyl-CoA and promotes histone acetylation at key inflammatory gene loci such as IL-1β and IL-6 (Langston et al, 2019; Lauterbach et al, 2019). Since glycerol supplementation increased acetyl-CoA levels in LPS-primed macrophages (Fig. 1D), we examined whether glycerol enhances chromatin acetylation at inflammatory cytokine promoters in LPS-primed pro-inflammatory macrophages. ChIP qPCR analysis of the IL-1β and IL-6 promoters using AcH3 antibodies, as described previously (Langston et al, 2019), was performed. Figure 4A shows that high concentrations of glycerol increased histone acetylation at the IL-1β and IL-6 gene promoters in LPS-primed pro-inflammatory macrophages. In this setting, inhibitors of AQP3, GK, and GPD2 suppressed glycerol-induced histone acetylation, indicating the involvement of glycerol-mediated G3P shuttle in histone acetylation. Consistent with these findings, glycerol supplementation for 24 h increased inflammatory cytokine mRNA levels, whereas inhibition of AQP3, GK, and GPD2 reduced these responses, including IL-1β and IL-6 (Fig. 4B). In contrast, glycerol at concentrations below 300 μM did not affect the expression of these genes (Fig. EV3C). In addition, incubation of M0-type BMDMs with glycerol did not affect the expression of

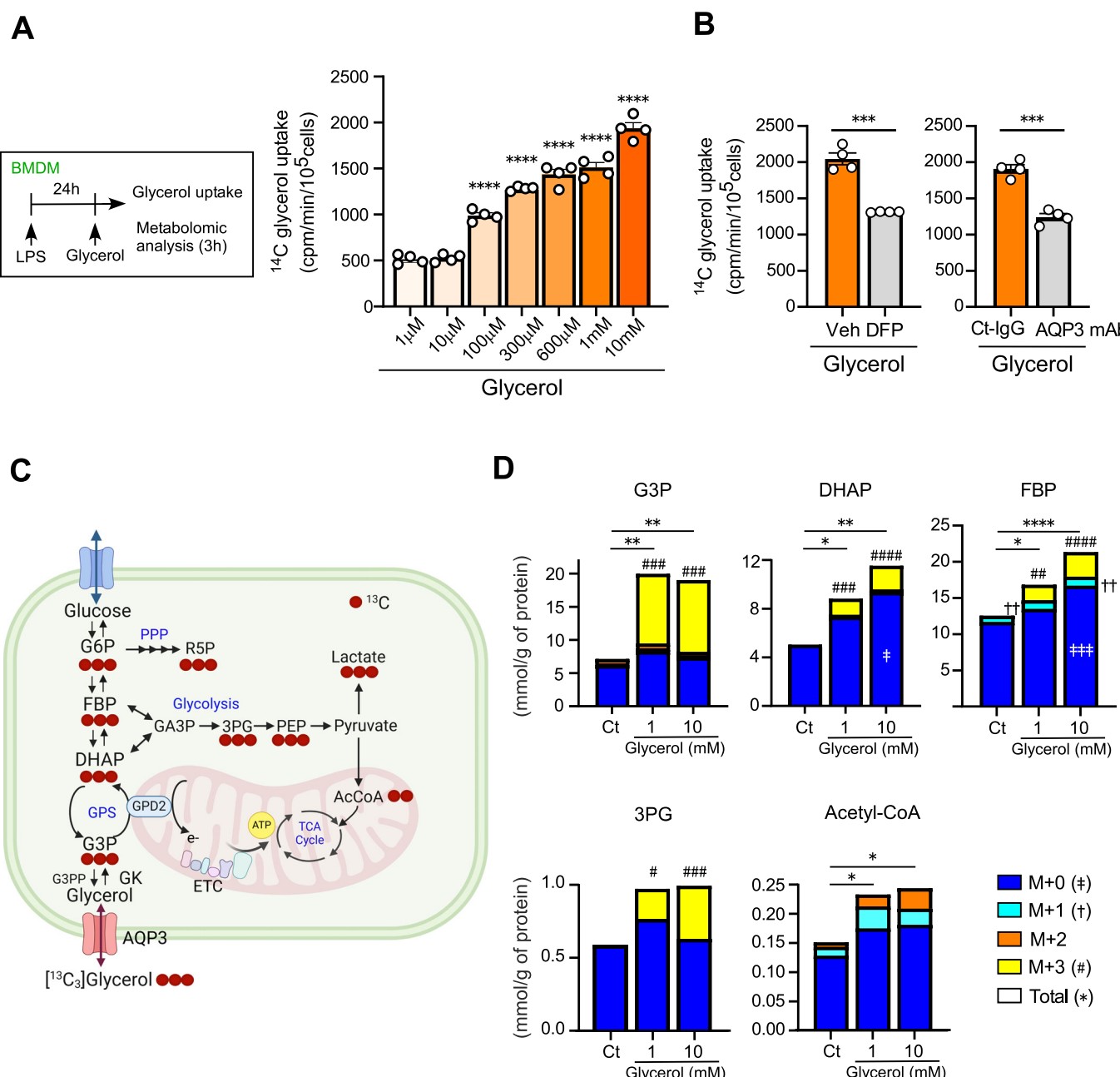

**Figure 1. Glycerol uptake and its metabolic pathway in pro-inflammatory macrophages.**

(A) Glycerol uptake was measured using [$^{14}$C]-labeled glycerol in BMDMs pretreated with LPS for 24 h (LPS-primed M1-type), following the addition of [$^{14}$C]-glycerol (1 µM to 10 mM) ($n = 4$). (****$P < 0.0001$ vs. 1 µM glycerol). (B) Glycerol uptake in the presence of DFP00173 (1 µM) or anti-AQP3 mAb (10 µg/ml) ($n = 4$) (DFP; ***$P = 0.0001$, AQP3 mAb; ***$P = 0.0001$). (C, D) Glycerol-starved LPS-primed BMDMs (24 h) were incubated with $^{13}$C$_3$ glycerol (1, 10 mM) for 3 h, and cellular metabolites were analyzed. (C) Atom mapping for $^{13}$C-glycerol tracing incorporation into G3P shuttle (GPS), glycolysis, and TCA cycle intermediates. Red circles represent $^{13}$C atoms. G6P glucose-6-phosphate, FBP fructose 1,6-bisphosphate, G3P glycerol-3-phosphate, DHAP dihydroxyacetone phosphate, GA3P DL-glyceraldehyde 3-phosphate, 3PG 3-phosphoglycerate, PEP phosphoenolpyruvate, R5P ribose 5-phosphate. (D) Total and $^{13}$C labeling amounts of metabolites in LPS-primed BMDMs ($n = 5$). (*$P < 0.05$, **$P < 0.01$, ****$P < 0.0001$ vs. control for the total amount. ‡$P < 0.05$, ‡‡‡$P < 0.001$ vs. control for M + 0 unlabeled amount, ††$P < 0.01$ vs. control for M + 1 amount. #$P < 0.05$, ##$P < 0.01$, ###$P < 0.001$, ####$P < 0.0001$ vs. control for M + 3 amount). Exact $P$ values: G3P for the total amount vs. control; 1 mM $P = 0.0031$, 10 mM $P = 0.0049$. G3P for M + 3 amount vs. control; 1 mM $P = 0.0003$, 10 mM $P = 0.0002$. DHAP for the total amount vs. control; 1 mM $P = 0.048$, 10 mM $P = 0.0017$. DHAP for M + 0 unlabeled amount vs. control; 10 mM $P = 0.0108$. DHAP for M + 3 amount vs. control; 1 mM $P = 0.0003$, 10 mM $P < 0.0001$. FBP for total amount vs. control; 1 mM $P = 0.019$, 10 mM $P < 0.0001$. FBP for M + 0 unlabeled amount vs. control; 10 mM $P = 0.0007$. FBP for M + 1 amount vs. control; 1 mM $P = 0.0042$, 10 mM $P = 0.0026$. FBP for M + 3 amount vs. control; 1 mM $P = 0.0012$, 10 mM $P < 0.0001$. 3PG for M + 3 amount vs. control; 1 mM $P = 0.0196$, 10 mM $P = 0.0004$. Acetyl-CoA for the total amount vs. control; 1 mM $P = 0.032$, 10 mM $P = 0.016$. Data information: All data presented in Fig. 1 are mean ± standard error of the mean (SEM). N values indicate biological replicates. (A, D) One-way ANOVA with Dunnett's multiple comparison test. (B) Unpaired t-tests. Source data are available online for this figure.

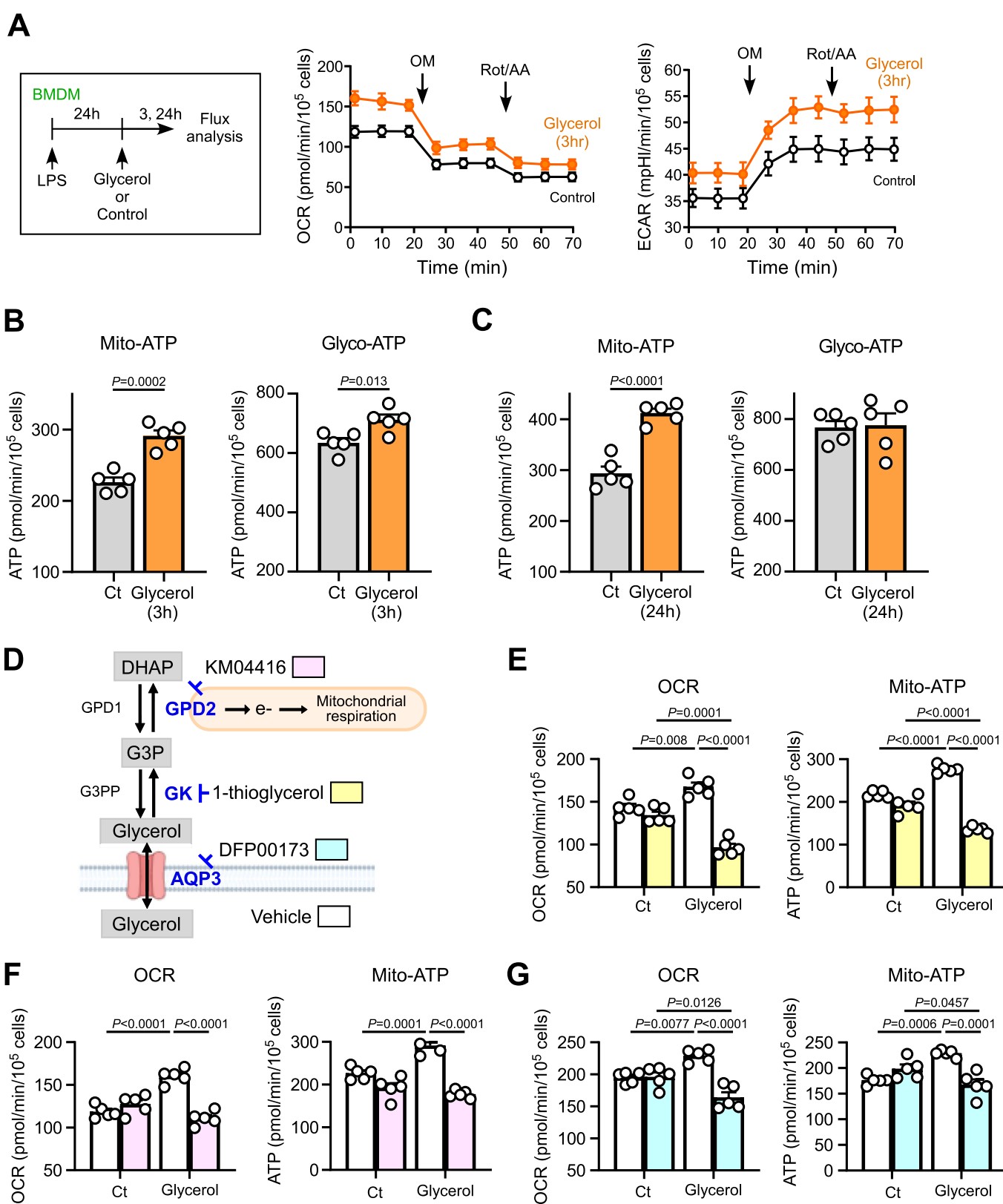

**Figure 2.** **High concentrations of extracellular glycerol induce mitochondrial ATP production via GPS.**

(A–C) Glycerol-starved LPS-primed BMDMs were incubated with glycerol (1 mM) or a glycerol-starved medium (serum-free, control). Cells were assessed using the ATP Rate Assay Kit with a Flux analyzer, as shown in (A) ($n = 5$). OM oligomycin, Rot/AA rotenone and antimycin A. (B, C) Mitochondrial- and glycolytic-ATP production after incubation with glycerol for 3 h (B) ($n = 5$) or for 24 h (C) ($n = 5$). (D) GPS pathway and inhibitors used. (E–G) Glycerol-starved LPS-primed BMDMs were treated with 1-thioglycerol (E), KM04416 (F), or DFP00173 (G) and subsequently incubated with glycerol (final 1 mM) for 3 h. Cellular metabolism was measured using the ATP Rate Assay Kit. (Left) OCR. (Right) Mitochondrial ATP production rate. ($n = 5$). Data information: All data presented in Fig. 2 are mean ± standard error of the mean (SEM). N values indicate biological replicates. (B, C) Unpaired t-tests. (E–G) Two-way ANOVA with Sidak's multiple comparisons test. Exact P values are reported, except where the adjusted P value was smaller than 0.0001, in which case it is reported as $P < 0.0001$. Source data are available online for this figure.

these mRNAs (Fig. EV3D). These findings suggest that glycerol supplementation promotes the histone acetylation of nuclear inflammatory cytokine genes, possibly through an increase in cellular acetyl-CoA levels, thereby enhancing the inflammatory response.

## AQP3 deficiency in macrophages attenuates HFD-induced obesity

Our experiments demonstrated that high extracellular glycerol affected cellular metabolism and promoted inflammatory responses in pro-inflammatory macrophages. Given the elevated serum glycerol levels observed in obesity (Maggs et al, 1995) (van der Merwe et al, 1999; van der Merwe et al, 2001), in which inflammatory macrophages play a key role in pathogenesis (Appari et al, 2018; Castoldi et al, 2015; Serbulea et al, 2018), we hypothesized that elevated levels of extracellular glycerol contribute to the pathology of obesity by affecting macrophage function during the progression of obesity. First, to determine whether adipocytes from obese mice secrete increased amounts of glycerol, we isolated adipocytes from the white adipose tissue (WAT) of high-fat diet (HFD)–fed obese and control lean wild-type mice and incubated them in the culture medium. We found that adipocytes from obese mice secreted significantly higher levels of glycerol and free fatty acids (FFAs) compared to those from lean mice, suggesting that extracellular glycerol concentrations are elevated in the context of obesity (Fig. EV4A,B).

To investigate the effects of glycerol on macrophages in vivo, we generated macrophage-specific AQP3 conditional knockout (cKO) mice to inhibit the influx of extracellular glycerol into macrophages. LysM-Cre[+]/AQP3[flox/flox] (AQP3 cKO) was established by crossing AQP3[flox/flox] mice with LysM-Cre mice, which selectively deleted target genes in myeloid lineage cells, including macrophages (Fig. EV4C). We confirmed AQP3 deletion in BMDMs from AQP3 cKO mice compared to AQP3[flox/flox]/LysM-Cre[−] control mice through DNA analysis (Fig. EV4D), qPCR (Fig. 4C), and immunoblotting (Fig. 4D; Appendix Fig. S1A). We showed that glycerol transport was impaired in AQP3 cKO BMDMs compared to that in control cells, with a reduction of ~50% (Fig. 4E). This also suggests that glycerol uptake by macrophages may be partially mediated by other channels such as AQP9.

We previously reported that macrophages from germline AQP3 knockout mice exhibit impaired M1 polarization in response to LPS, which we attributed in part to reduced $H_2O_2$ permeability through AQP3 (Hara-Chikuma et al, 2020). Similarly, BMDMs from AQP3 cKO mice showed suppressed M1 marker expression upon LPS stimulation (Fig. 4F). Flux analysis showed that macrophages from control mice showed increased ECAR and

glycolysis-derived ATP after 24-hour LPS stimulation. In contrast, AQP3 cKO macrophages attenuated this glycolytic shift, indicating an altered metabolic response during M1 polarization in response to LPS (Fig. 4G). As AQP3-deficient macrophages exhibit markedly altered responses to LPS and reduced M1 differentiation, we used pharmacological inhibition rather than genetic deletion in this study to assess the role of AQP3-mediated glycerol transport under more physiological conditions.

AQP3 cKO and control mice were fed either high fat diet (HFD) or normal chow (NC) for 6 weeks. Weight gain induced by HFD was significantly lower in AQP3 cKO mice than in control mice (Fig. 5A). There was no difference in HFD intake between the two groups (Fig. EV5A). HFD increased the WAT mass at various sites, including the epididymis and peritoneum, compared to NC-fed mice; this increase was attenuated in AQP3 cKO mice (Figs. 5B and EV5B). The HFD also increased the levels of obesity-associated serum components, including triglyceride (TG), FFAs, glycerol, fasting insulin, aspartate amino-transferase (AST), and alanine aminotransferase (ALT), as previously described (Hariri and Thibault, 2010), but these increases were suppressed in AQP3 cKO mice (Figs. 5C and EV5C). Glucose and insulin tolerance tests showed no significant differences between control and AQP3 cKO mice (Fig. EV5D,E). Analysis of mRNA expression in epididymal WAT revealed that HFD feeding significantly increased the expression of M1-type pro-inflammatory macrophage markers, including TNF-α, IL-1β, Nos2, CCL2, and CCR7, in control mice compared with lean controls (Fig. 5D). In contrast, this HFD-induced upregulation of pro-inflammatory macrophage markers was significantly attenuated in AQP3 cKO tissues, indicating that myeloid AQP3 contributes to obesity-associated macrophage polarization and inflammation in WAT. In other studies, M1-type pro-inflammatory macrophages have been found to predominate over M2-type macrophages in the WAT of obese subjects and mice, exacerbating obesity (Serbulea et al, 2018). Fluorescence-activated cell sorting (FACS) analysis revealed that HFD increased the proportion of CD11b[+] MHC II [high] CD206 [low] pro-inflammatory macrophages, whereas AQP3 deletion in macrophages significantly suppressed HFD-induced dominance of pro-inflammatory macrophages (Fig. 5E; Appendix Fig. S1B,C). Macrophages isolated from the WAT of HFD-fed mice showed an increased expression of pro-inflammatory cytokines, which were suppressed in AQP3 cKO macrophages (Fig. 5F). The expression levels of AQP3 and AQP9 were unchanged between the HFD and NC conditions, whereas glucose transporter 1 (GLUT1) was upregulated in macrophages from HFD-fed mice compared to control mice, as previously reported (Freemerman et al, 2014). These findings suggested that glycerol transport plays a role in the activation and inflammation of macrophages in the adipose tissue microenvironment during obesity development.

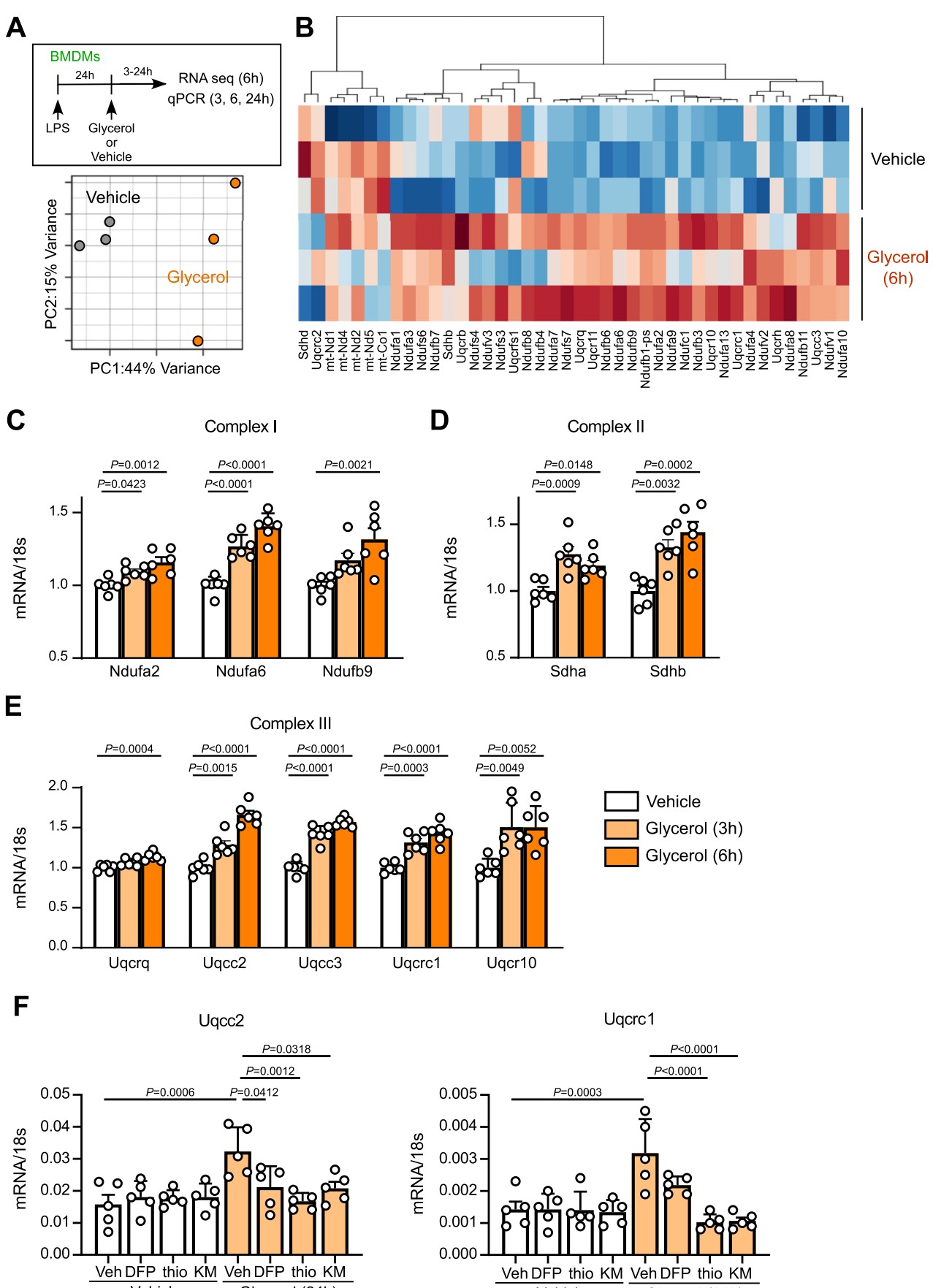

**Figure 3. Glycerol affects gene expressions related to mitochondrial respiration via GPS.**

(A, B) LPS-primed BMDMs were stimulated with glycerol for 6 h after glycerol starvation. RNA sequencing analysis was conducted ($n = 3$, biologically independent samples). (A) PCA of RNA sequencing data. (B) Heatmap showing DEGs categorized as "mitochondrial respiratory chain". All genes annotated to this pathway by RIAS analysis are displayed. (C–E) Gene expressions related to ETC Complex I, II, and III of the mitochondrial respiratory chain in LPS-primed BMDMs with or without glycerol (1 mM) for 3 or 6 h ($n = 6$). (F) Gene expression for $Uqcc2$ and $Uqcrc1$. LPS-primed BMDMs were treated with DFP00173 (DFP), 1-thioglycerol (thio), or KM04416 (KM) and subsequently incubated with glycerol (1 mM) for 24 h ($n = 5$). Data information: All data presented in Fig. 3 are mean ± standard error of the mean (SEM). N values indicate biological replicates. (C–E) One-way ANOVA with Dunnett's multiple comparison test. (F) One-way ANOVA with Tukey's multiple comparisons test. Exact $P$ values are reported, except where the adjusted $P$ value was smaller than 0.0001, in which case it is reported as $P < 0.0001$. Source data are available online for this figure.

# Discussion

In this study, we uncovered a previously unrecognized role for extracellular glycerol as a metabolic regulator of mitochondrial function and inflammatory response in pro-inflammatory macrophages. Metabolomics analysis with stable isotope tracing revealed that extracellular glycerol is rapidly metabolized into G3P by GK, and subsequently oxidized to DHAP by GPD2, thereby potentially donating electrons directly to the mitochondrial ETC (Langston et al, 2019; Liu et al, 2021; Oh et al, 2023; Oh et al, 2024). We further demonstrate that supplementation with high concentrations of glycerol significantly increased mitochondrial ATP production in LPS-primed pro-inflammatory macrophages. Importantly, this increase in ATP production was abolished by pharmacological inhibition of AQP3, GK, or mitochondrial GPD2, indicating that the glycerol-G3P-DHAP metabolic axis is critical for mitochondrial ATP generation via the G3P shuttle pathway. Additionally, the supplementation of a high concentration of glycerol upregulated the expression of genes encoding ETC complexes (Complexes I–III) in LPS-primed macrophages, whereas no such upregulation was observed in M0 macrophages. These results suggest that, in pro-inflammatory macrophages, elevated extracellular glycerol enhances mitochondrial ATP production through two complementary mechanisms: increased metabolic flux into the ETC via electron donation, and transcriptional activation of ETC-related genes.

The preferential upregulation of complex III–related genes by glycerol in LPS-primed macrophages is consistent with the metabolic architecture of the glycerol–G3P pathway under inflammatory conditions. In this pathway, GPD2 oxidizes G3P and donates electrons as $FADH_2$ directly to the ubiquinone pool, thereby bypassing complex I and feeding electron flux into complex III. In LPS-primed macrophages, inhibition of complex I reduces NADH-driven electron entry and increases reliance on alternative electron-donating routes (Jones and Divakaruni, 2020; Vercellino and Sazanov, 2022), so that electron flux originating from glycerol oxidation is expected to enter the ETC predominantly through the ubiquinone–complex III axis. This configuration likely imposes a greater respiratory demand on complex III and may drive the compensatory transcriptional upregulation of complex III subunits to sustain mitochondrial ATP production. Consistent with this mechanism, glycerol robustly induced ETC-related genes, particularly those associated with complex III, in LPS-primed macrophages, whereas M0 macrophages showed little or no transcriptional response.

Our data further suggest that the glycerol–G3P shuttle may link glycerol-driven mitochondrial bioenergetics to epigenetic regulation in pro-inflammatory macrophages. In LPS-primed macrophages, glycerol increased mitochondrial ATP production, elevated cellular acetyl-CoA levels, and enhanced histone H3 acetylation at

promoters of pro-inflammatory genes (e.g., $Il1b$ and $Il6$), whereas pharmacological inhibition of GPD2 attenuated these responses. One plausible explanation is that GPD2 inhibition blunts glycerol-dependent respiratory/ATP output and downstream citrate export, thereby limiting the nucleo-cytosolic acetyl-CoA supply required for histone acetylation. Changes in cytosolic redox state, including the shifts in pyruvate–lactate equilibrium, may further modulate these fluxes. At the same time, we note that reduced electron input into the ubiquinone pool upon GPD2 inhibition could, in principle, favor NADH oxidation through complex I and thereby increase mitochondrial $NAD^+$ regeneration, which might be expected to support PDH flux. How these potentially opposing effects are integrated across mitochondrial versus nucleo-cytosolic acetyl-CoA pools remains to be clarified. Moreover, because the GPD2 inhibitor used here (KM04416), an isothiazolinone derivative, may exhibit thiol/cysteine reactivity, off-target effects cannot be excluded (Burger et al, 2020). We therefore interpret the pharmacological data with appropriate caution, and future genetic validation of GPD2 and orthogonal chemical tools will be required to establish pathway specificity.

The physiological relevance of this glycerol-driven metabolic pathway was further supported by in vivo experiments in a mouse model of HFD-induced obesity. We found elevated serum glycerol levels accompanied by increased inflammatory activation of WAT macrophages in obese mice. Using macrophage-specific AQP3-deficient mice, we demonstrated that reduced glycerol uptake into macrophages significantly suppressed HFD-induced inflammatory gene expression both in WAT and in macrophages isolated from WAT. Although AQP3 cKO mice showed reduced weight gain under HFD, their food intake, glucose tolerance and insulin sensitivity were similar to those of control mice. This suggests that lower caloric intake or major alterations in systemic glucose homeostasis are not the primary causes of the reduced obese phenotype. Instead, the reduced weight gain coincided with a significant reduction of HFD-induced expression of pro-inflammatory macrophage markers in WAT, suggesting that the protection against obesity in AQP3 cKO mice is at least partly due to reduced inflammation in WAT rather than a direct effect on whole-body energy expenditure. These findings imply that elevated extracellular glycerol is not merely a passive metabolic byproduct of adipose lipolysis but rather an active immunometabolic signal that reinforces macrophage-driven inflammatory responses via the AQP3–G3P shuttle. However, we did not directly assess locomotor activity, energy expenditure, or brown/beige adipocyte activation in this study. Future work employing comprehensive metabolic phenotyping will be required to fully define the systemic metabolic consequences of macrophage AQP3 inhibition.

Although this study examined AQP3 as a representative glycerol channel, it is important to note that other aquaglyceroporins, such

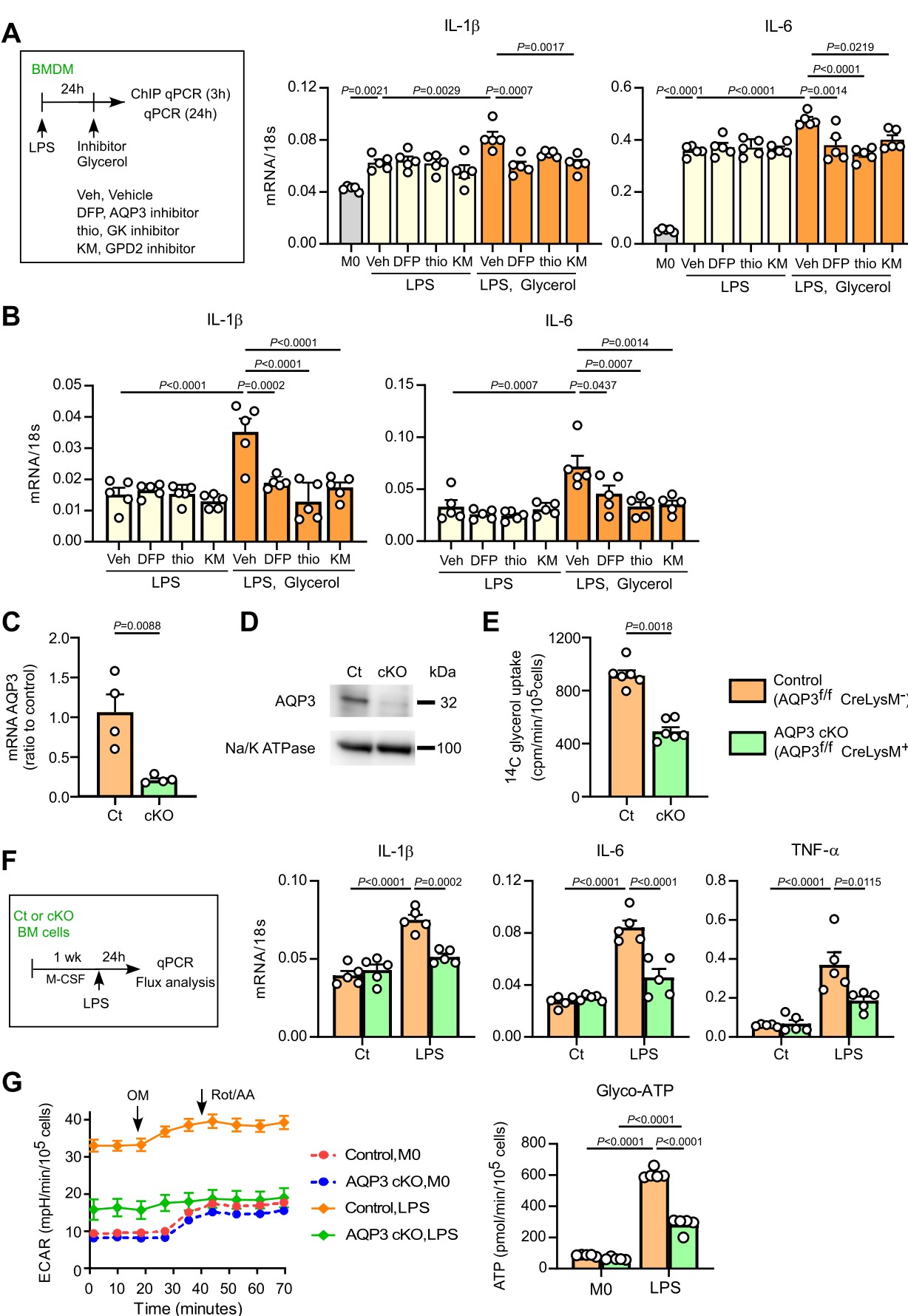

**Figure 4. Extracellular glycerol promotes histone acetylation for inflammatory cytokines *via* GPS.**

(A) LPS-primed BMDMs were treated with DFP00173 (DFP), 1-thioglycerol (thio), or KM04416 (KM), and subsequently incubated with glycerol (1 mM) for 3 h. ChIP qPCR analysis of histone acetylation with anti-AcH3 in *IL-1β* and *IL-6* promoter regions was performed ($n = 5$). (B) LPS-primed BMDMs were treated with DFP00173 (DFP), 1-thioglycerol (thio), or KM04416 (KM), and subsequently incubated with glycerol (1 mM) for 24 h. The indicated mRNA expression levels were quantified by qPCR ($n = 5$). (C, D) AQP3 expression in BMDMs from AQP3$^{fl/fl}$ LysM-Cre$^-$ control or AQP3$^{fl/fl}$ LysM-Cre$^+$ (AQP3 cKO) mice. (C) AQP3 mRNA expression by qPCR ($n = 4$ biologically independent samples). (D) Representative immunoblotting with anti-AQP3 and anti-Na/K ATPase as internal control from three independent experiments. (E–G) Bone marrow cells isolated from control and AQP3 cKO mice were differentiated into BMDMs by culturing for 1 week in the presence of M-CSF. The resulting BMDMs were stimulated with LPS (10 ng/mL) for 24 h. (E) Glycerol uptake by BMDMs ($n = 6$). (F) The indicated mRNA expression levels were quantified by qPCR ($n = 5$). (G) (left) Cells were assessed using the ATP Rate Assay Kit with a Flux analyzer. OM oligomycin, Rot/AA rotenone and antimycin A. (right) Glycolytic-ATP ($n = 5$). Data information: All data presented in Fig. 4 are mean ± standard error of the mean (SEM). *N* values indicate biological replicates. (A, B) One-way ANOVA with Tukey's multiple comparisons test. (C, E) Unpaired *t*-tests. (F, G) Two-way ANOVA with Sidak's multiple comparisons test. Exact *P* values are reported, except where the adjusted *P* value was smaller than 0.0001, in which case it is reported as $P < 0.0001$. Source data are available online for this figure.

as AQP9, are also capable of transporting glycerol to macrophages. Considering the functional similarities between AQP3 and AQP9, it is plausible that these transporters play overlapping or compensatory roles in glycerol uptake. For example, when AQP3 is inhibited or deleted, AQP9 may partially compensate for glycerol transport to sustain cellular metabolism. Conversely, AQP3 and AQP9 may act synergistically to facilitate glycerol uptake by macrophages with high metabolic demands, such as during LPS stimulation or in inflammatory environments. However, the relative contributions of AQP3 and AQP9 to glycerol uptake and their potential cooperation remain unclear and have not been addressed in the current study. Future studies employing genetic deletion models (e.g., AQP3 and AQP9 double-knockout macrophages) or specific inhibitors targeting both transporters could provide further insights into their individual and combined roles in macrophage metabolism and inflammation.

In summary, our findings highlight the role of glycerol in cellular energy metabolism and the inflammatory response of pro-inflammatory macrophages, particularly in environments with high extracellular glycerol concentrations, such as obese adipose tissue. These findings may inform new strategies for modulating inflammation by targeting glycerol metabolism in pro-inflammatory macrophages.

# Methods

### Reagents and tools table

| Reagent/resource | Reference or source | Identifier or catalog number |
|---|---|---|
| **Experimental models** | | |
| AQP3-flox | Cyagen Biosciences Inc. | N/A |
| C57BL/6J | Japan SLC | N/A |
| B6.129P2-Lyzs<tm1(cre)Ifo> | Riken Bio Resource Center | RBRC02302 |
| **Recombinant DNA** | | |
| Adeno-associated virus for mouse glycerol kinase | Origine | MR208395A1V |
| AAV2-CMV-GFP Control Particle | Origine | CV900001S |
| **Antibodies** | | |
| CD45 Monoclonal Antibody (30-F11), eFluor™ 450 | Thermo Fisher Scientific | Cat# 48-0451-82, RRID:AB_1518806 |
| CD11b Monoclonal Antibody (M1/70), FITC | Thermo Fisher Scientific | Cat# 11-0112-82, RRID:AB_464935 |

| Reagent/resource | Reference or source | Identifier or catalog number |
|---|---|---|
| F4/80 Monoclonal Antibody (BM8), APC | Thermo Fisher Scientific | Cat# 17-4801-82, RRID:AB_2784648 |
| MHC Class II (I-A/I-E) Monoclonal Antibody (M5/114.15.2), PE-Cyanine7 | Thermo Fisher Scientific | Cat# 25-5321-82, RRID:AB_10870792 |
| CD206 (MMR) Monoclonal Antibody (MR6F3), PE | Thermo Fisher Scientific | Cat# 12-2061-82, RRID:AB_2637422 |
| Anti-acetyl-Histone H3 | Millipore | Cat# 06-599, RRID:AB_2115283 |
| Normal Rabbit IgG | Cell Signaling Technology | Cat# 2729 |
| Anti-Aquaporin 3 Antibody | Millipore | Cat# AB3276-200UL, RRID:AB_2059552 |
| Anti-Na + /K+ ATPase α-1 Antibody, clone C464.6 | Millipore | Millipore Cat# 05-369, RRID:AB_309699 |
| Anti-mouse IgG, HRP-linked Antibody | Cell Signaling Technology | Cat# 7076 |
| Anti-rabbit IgG, HRP-linked Antibody | Cell Signaling Technology | Cat# 7074 |
| **Oligonucleotides and other sequence-based reagents** | | |
| 18 s Forward primer:GAGGCCCTGTAATTGGAATGAG | This paper | N/A |
| 18 s Reverse primer:GCAGCAACTACTTTAATATACGC-TATTGG | This paper | N/A |
| TNF-α Forward primer:CCCTCACACTCAGATCATCTTCT | This paper | N/A |
| TNF-α Reverse primer:GCTACGACGTGGGCTACAG | This paper | N/A |
| IL-6 Forward primer:TAGTCCTTCCTACCCCAATTTCC | This paper | N/A |
| IL-6 Reverse primer:TTGGTCCTTAGCCACTCCTTC | This paper | N/A |
| IL-1β Forward primer:GCAACTGTTCCTGAACTCAACT | This paper | N/A |
| IL-1β Reverse primer:ATCTTTTGGGGTCCGTCAACT | This paper | N/A |
| IFN-γ Forward primer:GAACTGGCAAAAGGATGGTGA | This paper | N/A |
| IFN-γ Reverse primer:TGTGGGTTGTTGACCTCAAAC | This paper | N/A |
| AQP3 Forward primer:TCTTTGACCAGTTCATAGGCAC | This paper | N/A |
| AQP3 Reverse primer:GGCAGGGTTGACGGCATAG | This paper | N/A |
| Uqcc2 Forward primer:TCCGGGAGGGAGAGAACAC | This paper | N/A |
| Uqcc2 Reverse primer:AGGGTACTTGTGCTTGTAGTAGT | This paper | N/A |

| Reagent/resource | Reference or source | Identifier or catalog number |
|---|---|---|
| Uqcc3 Forward primer:TGGCTCGTAAAGCACTTGTGG | This paper | N/A |
| Uqcc3 Reverse primer:CATCCCCGCTAACTGTCCA | This paper | N/A |
| Uqcr10 Forward primer:CGAGCGAGCCTTCGATCAG | This paper | N/A |
| Uqcr10 Reverse primer:ACAGTTTCCCCTCGTTGATGT | This paper | N/A |
| Uqcrc1 Forward primer:AGACCCAGGTCAGCATCTTG | This paper | N/A |
| Uqcrc1 Reverse primer:GCCGATTCTTTGTTCCCTTGA | This paper | N/A |
| Uqcrq Forward primer:CCTACAGCTTGTCGCCCTTT | This paper | N/A |
| Uqcrq Reverse primer:GATCAGGTAGACCACTACAAACG | This paper | N/A |
| Ndufa2 Forward primer:TTGCGTGAGATTCGCGTTCA | This paper | N/A |
| Ndufa2 Reverse primer:ATTCGCGGATCAGAATGGGC | This paper | N/A |
| Ndufa6 Forward primer:TCGGTGAAGCCCATTTTCAGT | This paper | N/A |
| Ndufa6 Reverse primer:CTCGGACTTTATCCCGTCCTT | This paper | N/A |
| Ndufb9 Forward primer:GGTACTTTGCTTGCTTGATGAGA | This paper | N/A |
| Ndufb9 Reverse primer:TGGGAAGATATACGGCTGAGG | This paper | N/A |
| Sdha Forward primer:GGAACACTCCAAAAACAGACCT | This paper | N/A |
| Sdha Reverse primer:CCACCACTGGGTATTGAGTAGAA | This paper | N/A |
| Sdhb Forward primer:AATTTGCCATTTACCGATGGGA | This paper | N/A |
| Sdhb Reverse primer:AGCATCCAACACCATAGGTCC | This paper | N/A |
| Glycerol kinase Forward primer:TGAAGAAAGCGAAATCCGTTACT | This paper | N/A |
| glycerol kinase Reverse primer:CCCAAAGGCAGACTACAGAAG | This paper | N/A |
| Glut1 Forward primer:CAGTTCGGCTATAACACTGGTG | This paper | N/A |
| Glut1 Reverse primer:GCCCCCGACAGAGAAGATG | This paper | N/A |
| Nos2 Forward primer:GTTCTCAGCCCAACAATACAAGA | This paper | N/A |
| Nos2 Reverse primer:GTGGACGGGTCGATGTCAC | This paper | N/A |
| Ccl2 Forward primer:GTGATGGAGGGGGTCAGGA | This paper | N/A |
| Ccl2 Reverse primer:GGGATGGGACAGCCTAAACT | This paper | N/A |
| Ccr7 Forward primer:TGTACGAGTCGGTGTGCTTC | This paper | N/A |
| Ccr7 Reverse primer:GGTAGGTATCCGTCATGGTCTTG | This paper | N/A |
| CD206 Forward primer:GTTCACCTGGAGTGATGGTTCTC | This paper | N/A |
| CD206 Reverse primer:AGGACATGCCAGGGTCACCTTT | This paper | N/A |
| CD68 Forward primer:GTGTCTGATCTTGCTAGGACC | This paper | N/A |
| CD68 Reverse primer:TGTGCTTTCTGTGGCTGTAG | This paper | N/A |
| Cxcl1 Forward primer:CTGGGATTCACCTCAAGAACATC | This paper | N/A |

| Reagent/resource | Reference or source | Identifier or catalog number |
|---|---|---|
| Cxcl1 Reverse primer:CAGGGTCAAGGCAAGCCTC | This paper | N/A |
| IL-6 Forward primer for Chip qPCR :AGGAGTGTGAGGCAGAGAGC | This paper | N/A |
| IL-6 Reverse primer for Chip qPCR:GTCTCCTGCGTGGAGAAAAG | This paper | N/A |
| IL-1β Forward primer for Chip qPCR:ATGTGCGGAACAAAGGTAGG | This paper | N/A |
| IL-1β Reverse primer for Chip qPCR:CCTGACCCACACAAGGAAGT | This paper | N/A |
| **Chemicals, enzymes and other reagents** | | |
| [1,3-$^{14}$C] glycerol | ARC | ARC0336A |
| [1,3-$^{14}$C] glycerol | Perkin Elmer | NEC441x |
| Glycerol-$^{13}$C3 | Toronto Research Chemicals | Cat# TRC-G598403 |
| D-[6-3H(N)]-glucose | Perkin Elmer | NET100C |
| Glycerol | FUJIFILM Wako Pure Chemical Corporation | Cat# 075-00616 |
| Lipopolysaccharides from Salmonella enterica serotype typhimurium | Sigma-Aldrich | Cat# L6143 |
| Mouse IFN-gamma, Animal-Free Recombinant Protein | Thermo Fisher Scientific | Cat# AF-315-05 |
| KM04416 | MedChemExpress | Cat# HY-148685 |
| 1-Thioglycerol | Sigma-Aldrich | Cat# M6145 |
| SB204990 | Selleck Biotechnology | Cat# E1287 |
| DFP00173 | MedChemExpress | Cat# HY-126073 |
| TB Green® Premix Ex Taq™ II | TAKARA BIO | Cat# RR820 |
| One Step TB Green® PrimeScript™ PLUS RT-PCR Kit | TAKARA BIO | Cat# RR096 |
| TRIzol™ Reagent | Thermo Fisher Scientific | Cat# 15596018 |
| Rodent Diet With 60 kcal% Fat | Research Diets | Cat# D12492 |
| Methanol | FUJIFILM Wako Pure Chemical Corporation | Cat# 134-14523 |
| Ultrapure Water | FUJIFILM Wako Pure Chemical Corporation | Cat# 210-01303 |
| D(-)-Mannitol | FUJIFILM Wako Pure Chemical Corporation | Cat# 137-00843 |
| Chloroform | FUJIFILM Wako Pure Chemical Corporation | Cat# 036-01926 |
| 2-Morpholinoethanesulfonic acid | DOJINDO LABORATORIES | Cat# 341-01622 |
| L-Methionine Sulfone | FUJIFILM Wako Pure Chemical Corporation | Cat# 502-76641 |
| D-Camphor-10-sulfonic acid sodium salt | FUJIFILM Wako Pure Chemical Corporation | Cat# 037-01032 |
| 3-aminopyrrolidine | Sigma-Aldrich | Cat#F024526 |
| Trimesate | FUJIFILM Wako Pure Chemical Corporation | Cat#QF-7219 |
| UltrafreeMC-PLHCC 250/pk for Metabolome Analysis (5 kDa) | Human metabolome technologies | Cat# UFC3LCCNB-HMT |
| D(+)-Glucose | FUJIFILM Wako Pure Chemical Corporation | Cat#043-31165 |
| RPMI 1640 with L-Gln and HEPES | NACALAI TESQUE | Cat# 30263-95 |
| Recombinant Mouse M-CSF | BioLegend | Cat# 576404 |

| Reagent/resource | Reference or source | Identifier or catalog number |
|---|---|---|
| 2-Mercaptoethanol | NACALAI TESQUE | Cat# 21438-82 |
| cOmplete Protease Inhibitor Cocktail | Roche | Cat# 11836145001 |
| RIPA Buffer (10X) | Cell Signaling Technology | Cat# 9806 |
| NuPAGE Bis-Tris Mini Protein Gels, 10%, 1.0 mm | Invitrogen | Cat# NP0302BOX |
| Trans-Blot Turbo Mini 0.2 μm PVDF Transfer Packs | Bio-Rad | Cat# 1704156 |
| Immobilon Western Chemiluminescent HRP Substrate | Millipore | Cat# WBKLS0100 |
| Insulin, Human, recombinant, Animal-derived-free | FUJIFILM Wako Pure Chemical Corporation | Cat# 090-06481 |
| Seahorse XF Real-Time ATP Rate Assay Kit | Agilent Technologies | Cat# 103592-100 |
| SimpleChIP® Enzymatic Chromatin IP Kit | Cell Signaling Technology | Cat# 9003 |
| RNeasy Mini Kit | QIAGEN | Cat# 74104 |
| PrimeScript™ RT reagent Kit | TAKARA BIO | Cat# RR037 |
| CellAmp™ Direct RNA Prep Kit for RT-PCR | TAKARA BIO | Cat# 3732 |
| High Sensitivity Free Glycerol Fluorometric Assay Kit | Sigma-Aldrich | Cat# MAK270 |
| LabAssay (TM) NEFA | FUJIFILM Wako Pure Chemical Corporation | Cat# 633-52001 |
| LabAssay (TM) Glucose | FUJIFILM Wako Pure Chemical Corporation | Cat# 638-50971 |
| L-Type CHO-M Enzyme Color A | FUJIFILM Wako Pure Chemical Corporation | Cat# 460-11892 |
| TG-M test wako | WAKO | Cat# 464-89001 |
| NEFA C-test Wako | WAKO | Cat# 279-75401 |
| L-Type Wako LD· IF | WAKO | Cat # 460-88001 |
| L-Type Wako AST· J2 | WAKO | Cat # 464-87801 |
| L-Type Wako ALT· J2 | WAKO | Cat # 460-87901 |
| L-Type ALP IFCC | WAKO | Cat # 466-88101 |
| Mouse/Rat Insulin Assay Kit | TAKARA BIO | Cat# MS1108 |
| Glucose Uptake Assay Kit | DOJINDO | Cat# UP02 |
| Protein Assay Reagent A | BIO RAD | Cat# 5000113 |
| Protein Assay Reagent B | BIO RAD | Cat# 5000114 |
| Fetal bovine serum | HyClone | SH30088.03 |
| Fetal bovine serum | Sigma-Aldrich | Cat# 172012 |
| Fetal bovine serum | Sigma-Aldrich | Cat# 173012 |
| Slide-A-Lyzer Dialysis Cassette | Thermo Fisher Scientific | Cat# 66810 |
| Bovine serum albumin | Sigma-Aldrich | Cat# A7030-50G |
| Formaldehyde solution | NACALAI TESQUE | Cat# 16223-55 |
| **Software** | | |
| FLowJo | FlowJp LLC | https://www.flowjo.com/solutions/flowjo |
| GraphPad Prism8 | GraphPad Inc | https://www.graphpad.com/scientific-software/prism/ |
| Agilent Seahorse XF Wave | Agilent | N/A |
| MasterHands | Human Metabolome Technologies Inc. | N/A |
| BioRender | BioRender Inc. | https://app.biorender.com/ |

| Reagent/resource | Reference or source | Identifier or catalog number |
|---|---|---|
| RIAS Omics Analysis System | Rhelixa | https://rias.rhelixa.com/ |
| **Other** | | |
| NIPRO FreeStyle Freedom Lite | NIPRO | Cat# 30854000 |
| Blood Glucose Monitoring System | NIPRO | Cat# 30221003 |
| High-fat diet | Research Diets | D12492 |
| CytoFlex | Beckman Coulter | N/A |
| CFX Opus Real-time PCR System | Bio-Rad Laboratories | N/A |
| Agilent G7100A CE System (CE-TOFMS) | Agilent Technologies | N/A |
| Agilent 6530 Q-TOF LC/MS System | Agilent Technologies | N/A |
| XF24 Extracellular flux analyzer | Seahorse Biosciences | N/A |

## Mice

All animal experiments were approved by the President of Keio University, following the consideration by the Institutional Animal Care and Use Committee of Keio University (Approval no: 16075) and by Genetic Modification Safety Committee, Keio University School of Medicine (approval no. 28-029), and were carried out in accordance with institutional procedures, national guidelines, and the relevant national laws on the protection of animals.

C57BL/6 mice were purchased from Japan SLC, Inc. Mice containing loxP-flanked AQP3 alleles (AQP3$^{fl/fl}$) were created by Cyagen (USA) through ES genome engineering. B6.129P2-Lyzs < tm1(cre)Ifo > (LysM-Cre) mice (Clausen et al, 1999) were provided by the Riken Bio Resource Center (RBRC02302). AQP3$^{fl/fl}$ mice were crossed with LysM-Cre mice to generate AQP3$^{fl/fl}$ LysM-Cre$^{+/-}$ mice.

All mice were maintained in an SPF animal facility at Keio University under standard housing conditions (22 ± 2 °C, 40–60% humidity, 12-h light/12-h dark cycle) with free access to standard chow and water. Mice were group-housed (3–5 animals/cage) on autoclaved bedding with environmental enrichment.

## Bone marrow-derived macrophage preparation

Single cell suspensions of bone marrow cells were collected from femur and tibia, and cultured in RPMI 1640 (Invitrogen) containing 10 ng/ml M-CSF (R&D Systems, Inc.), 10% fetal bovine serum, 50 μM 2-mercaptoethanol, 2 mM L-glutamine, 25 mM HEPES, 1 mM nonessential amino acids, 1 mM sodium pyruvate, 1% penicillin-streptomycin for at least 6 days. More than 90% of cultured cells were confirmed as macrophages by FACS analysis. M0 macrophages (BMDMs) were stimulated with LPS (10 ng/ml, Sigma-Aldrich). In case, BMDMs were incubated with LPS (1 ng/ml) and IFN-γ (10 ng/ml, R&D) to differentiate into pro-inflammatory macrophages.

Polarized M1 macrophages were infected with recombinant adeno-associated virus for mouse glycerol kinase (Origine, MR208395A1V) or with control adeno-associated particles (Origine).

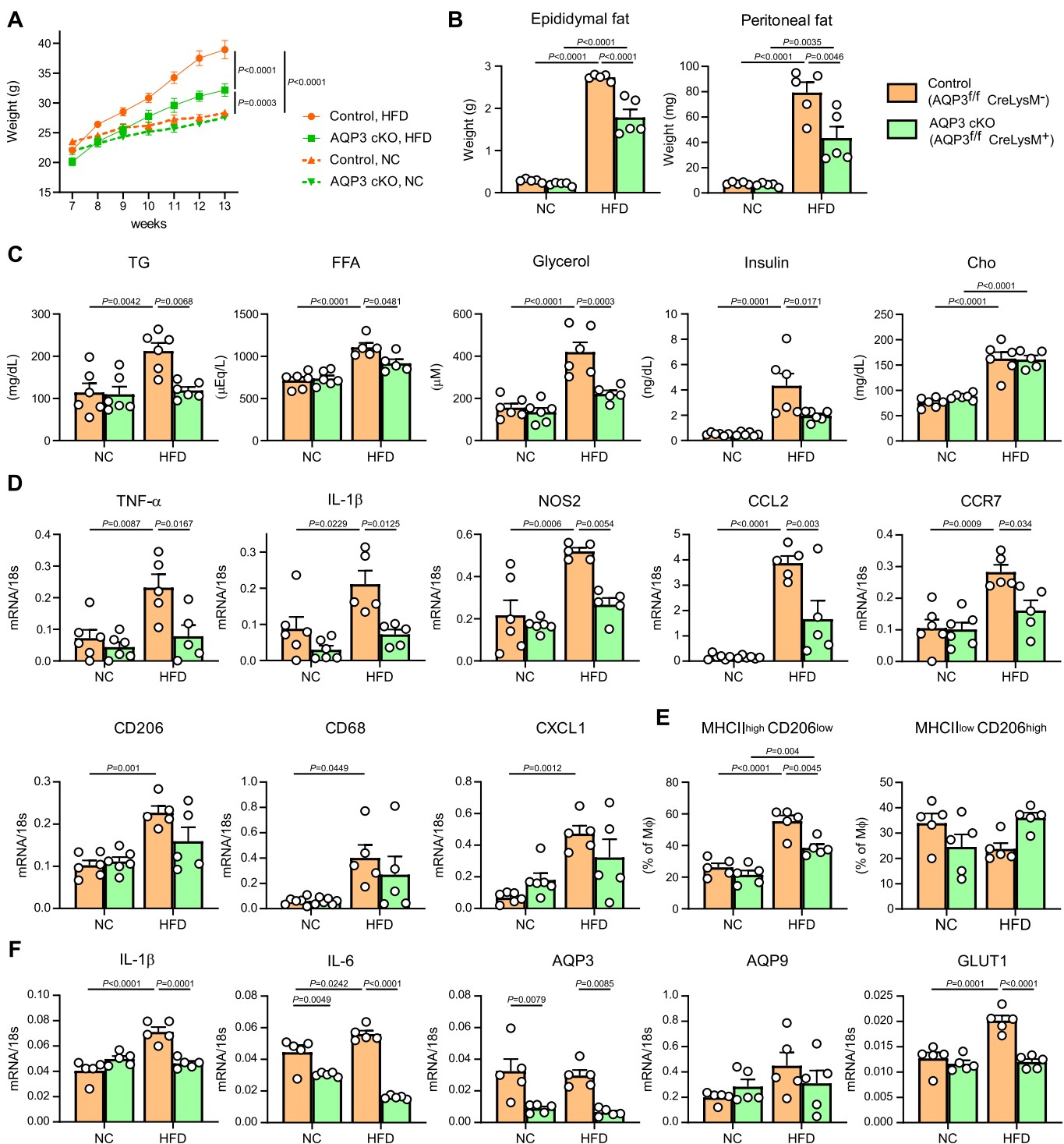

**Figure 5. AQP3 deficiency in macrophages attenuates HFD-induced obesity.**

(A–F) AQP3 cKO and control mice were fed HFD or normal chow (NC) for 6 weeks. (A) Weights were measured weekly ($n = 5$). (B) Weights of adipose tissue from the epididymal and peritoneal sites ($n = 5$). (C) Serum triglyceride (TG), non-esterified fatty acid (FFA), glycerol, fasting insulin, and cholesterol (Cho) contents ($n = 6$). (D) Expression of indicated mRNAs using qPCR in epididymal adipose tissues. Data are expressed as the ratio of 18 s RNA ($n = 5$-6). (E) Dispersed cells from the WAT were analyzed using FACS. (left) % of CD11b$^+$ F4/80$^+$ MHCII $^{high}$ CD206 $^{low}$ M1-type macrophages among CD45$^+$ immune cells. (right) % of CD11b$^+$ F4/80$^+$ MHC II $^{low}$ CD206 $^{high}$ M2-type macrophages among CD45$^+$ immune cells ($n = 5$-6). (F) Expression of mRNAs encoding IL-1β, IL-6, AQP3, AQP9, and GLUT1 using qPCR in macrophages isolated from WAT. Data were expressed as the ratio of 18 s RNA ($n = 5$). Data information: All data presented in Fig. 5 are mean ± standard error of the mean (SEM). N values indicate biological replicates. (A–F) Two-way ANOVA with Sidak's multiple comparisons test. Exact P values are reported, except where the adjusted P value was smaller than 0.0001, in which case it is reported as $P < 0.0001$. Source data are available online for this figure.

## Reagents

An anti-mouse AQP3 monoclonal antibody was developed as described previously (Hara-Chikuma et al, 2020). DFP00173 and KM04416 were purchased from MedChemExpress (Monmouth Junction, NJ, USA). 1-thioglycerol was purchased from Sigma-Aldrich. SB204990 was purchased from Selleck Biotech.

## Glycerol and glucose uptake assay

Cells were incubated with [1,3-$^{14}$C]-glycerol (ARC, ARC0336A, or Perkin Elmer, NEC441x) or D-[6-$^3$H(N)]-glucose (Perkin Elmer, NET100C) in RPMI for 3 min at room temperature. After washing cells three times with ice-cold PBS, cells were disrupted with 1 M NaOH. Cell-associated radioactivity was determined by scintillation counting.

Glucose uptake was also determined using the Glucose Uptake Assay Kit according to the manufacturer's protocol (DOJINDO, UP02).

## RNA extraction and real-time quantitative RT-PCR

Total RNA was extracted using the RNeasy Mini kit according to the manufacturer's instructions (QIAGEN). The cDNA was reverse transcribed from total RNA using the Prime Script RT reagent kit (Takara Bio, Otsu, Japan). Quantitative RT-PCR was performed using SYBR Green I (Takara Bio) and CFX Opus Real-time PCR System (Bio-Rad).

## RNA sequence analysis

The extracted RNAs using the RNeasy Mini kit were subjected to RNA-seq, which was outsourced to Rhelixa Inc. (Tokyo, Japan). In brief, cDNA library preparation was performed with the NEBNext Poly(A) mRNA Magnetic Isolation Module and NEBNext Ultra II Directional RNA Library Prep Kit. The Illumina NovaSeq 6000 system was used for sequencing in the 150-base paired-end mode. FastQC v.0.11.7 was used for a quality check of the sequencing run, and Trimmomatic v.0.38 was used to trim the sequenced reads. HISAT2 v.2.1.0. was used to map the sequenced reads to the mice reference genome sequences (mm10). The raw read counts were normalized with transcripts per million (TPM). Principal component analysis (PCA) of the normalized counts was conducted, and each sample was projected onto the 2D plane of the first and second PCA axes using stats (Version 3.6.1) and gplots (Version 3.0.1.1) R packages. We used the complete DESeq2 results table and defined DEGs as genes with valid Entrez Gene IDs and a Benjamini–Hochberg adjusted *P* value (*p*adj) <0.05, unless otherwise specified. Normalized read counts and corresponding *Z*-scores were calculated by the sequencing service provider (Rhelixa Inc.) using the stats (R v3.6.1) and gplots (v3.0.1.1) packages in R. Heatmaps were then generated from these Z-scores using RIAS (Rhelixa Inc.).

## Metabolome analysis

Metabolome analysis was performed as previously described (Mizota et al, 2022; Tanosaki et al, 2020). Briefly, LPS-primed BMDMs (24 h) were incubated with RPMI 1640 containing $^{13}$C$_3$-labeled glycerol (Tronto Research Chemical, G598403, 1 or 10 mM)

for 3 h. The cells were washed with 5% mannitol, and metabolites were extracted with a solvent composed of methanol, chloroform, 200 μM 2-Morpholinoethanesulfonic acid, 200 μM L-methionine sulfone, and 1 μM D-Camphor-10-sulfonic Acid. After centrifugation at 10,000×*g* at 4 °C for 15 min, the water and methanol layers were filtered through the 5 kDa cut-off filter (UFC3LCCNB, Human metabolome technologies). Then, the filtrate was lyophilized and dissolved in deionized water containing reference compounds (200 μM each of 3-aminopyrrolidine and trimesate). Metabolites were analyzed with the Agilent CE time-of-flight MS (CE-TOFMS) system equipped with the Agilent G7100A CE instrument, and the Agilent 6530 Q-TOF LC/MS system (Agilent Technologies). Raw data were processed using MasterHands. The metabolites were identified by matching m/z and corrected migration times with those in our standard library. The absolute concentration was quantified based on the ratio among peak areas of each metabolite, internal, and external standard compounds.

## Measurement of oxygen consumption rate and acidification rate

Cellular energy metabolism of macrophages was measured using an XF24 Extracellular flux analyzer (Seahorse Biosciences, North Billerica, MA, USA). Briefly, BMDMs or LPS-stimulated macrophages seeded on XF24 cell culture microplates were measured for OCR and ECAR using the XFe ATP Rate Assay Kit by sequential addition of oligomycin (OM, 1.5 μM) and rotenone/antimycin A (1.5 μM). ATP production rates were calculated according to the manufacturer's instructions using the method described by Desousa et al (Desousa et al, 2023).

## Chromatin immunoprecipitation (ChIP) assay

Acetylation at inflammatory gene promoters was measured by ChIP assay. LPS-primed BMDMs (24 h) were treated with inhibitors and subsequently stimulated with or without glycerol (3 h). ChIP assay was performed with SimpleChiP Rnzymatic Chromatin IP Kit (CST, #9003) according to the manufacturer's protocol. The immunoprecipitation was performed with anti-acetyl-histone H3 (#06-599; Merck Millipore) or IgG (#2729, CST). Fold enrichment was calculated as ChIP signals normalized to input.

## Immunoblotting

Cells were lysed with membrane extraction buffer (HEPES pH 7.4, 250 mM sucrose, 1 mM EDTA, 1 mM EGTA, 1% protease inhibitor cocktail) for membrane protein analysis. The supernatant (10,000×*g*, 10 min, 4 °C) was used for immunoblotting with AQP3 (Millipore) and Na$^+$/K$^+$ ATPase (Millipore).

## High-fat diet-induced mouse model

To induce obesity, male mice were fed a high-fat diet (HFD; Research Diets D12492, 60% calories from fat) at 6–7 weeks of age for 4–6 weeks. Epididymal, peritoneal, and retroperitoneal fat tissues were isolated and weighed. Serum components were measured with the following kits: triglyceride (TG-M test, Wako), free fatty acid (NEFA C-test, Wako), glycerol (High Sensitivity Free

Glycerol Assay kit, Sigma-Aldrich), cholesterol (L-type WAKO CHO-M, Wako) and Insulin (Insulin ELISA kit, Morinaga). Total RNA was extracted from adipocyte tissues isolated from epididymal fat using the RNeasy Mini kit as described above. Macrophages were isolated from epididymal fat by the method described previously (Orr et al, 2013).

## Glucose tolerance test

Mice were fasted overnight with ad libitum access to water. Fasting blood glucose levels and body weight were measured, followed by intraperitoneal injection of D-glucose solution (0.1 g/ml saline, 0.1 mL per 10 g body weight). Blood glucose levels were measured every 30 min post-injection using a blood glucose meter (FreeStyle Freedom Lite, NIPRO).

## Insulin tolerance test

Mice were fasted for 3 h with ad libitum access to water. Fasting blood glucose levels and body weight were measured, followed by intraperitoneal injection of insulin solution (1 U per kg body weight). Blood glucose levels were measured every 15 min post-injection using a blood glucose meter (FreeStyle Freedom Lite, NIPRO).

## Statistical analysis

Statistical analysis was performed using the two-tailed Student's $t$-test, one-way, or two-way ANOVA as indicated using GraphPad Prism8 (San Diego, CA, USA). Exact $P$ values are reported, except where the adjusted $P$ value was smaller than 0.0001, in which case it is reported as $P < 0.0001$.

# Data availability

· Raw RNA-seq data have been deposited in the DDBJ Sequence Read Archive under BioProject accession number PRJDB39611: https://ddbj.nig.ac.jp/search/entry/bioproject/PRJDB39611.
· RNA-seq data have been deposited in the Gene Expression Archive (GEA) under accession E-GEAD-1180: https://ddbj.nig.ac.jp/public/ddbj_database/gea/experiment/E-GEAD-1000/E-GEAD-1180.

The source data of this paper are collected in the following database record: biostudies:S-SCDT-10_1038-S44319-026-00747-y.

# Peer review information

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

## Acknowledgements

We thank Dr. Irmgard Foerster (Life and Medical Sciences Institute, University of Bonn) for providing CreLysM mice. We thank Mr. Gen Itai for the generation of genome-edited mice at the Laboratory Animal Center, Keio University School of Medicine. We thank Dr. Makoto Suematsu, Mr. Inaba and Mr. Koda for their helpful discussion.

This work was supported by grants from Research Ministry of Education, Culture, Sports, Science (21K06974, MH-C) and Keio University Academic Development Funds (MH-C).

## Author contributions

**Manami Tanaka**: Validation; Investigation; Visualization; Methodology. **Takako Hishiki**: Formal analysis; Investigation; Methodology. **Tomomi Matsuura**: Formal analysis; Investigation; Methodology. **Masato Yasui**: Supervision. **Shunsuke Chikuma**: Data curation; Formal analysis; Validation; Methodology. **Mariko Hara-Chikuma**: Conceptualization; Resources; Data curation; Formal analysis; Supervision; Funding acquisition; Validation; Investigation;

Visualization; Methodology; Writing—original draft; Project administration; Writing—review and editing.

Source data underlying figure panels in this paper may have individual authorship assigned. Where available, figure panel/source data authorship is listed in the following database record: biostudies:S-SCDT-10_1038-S44319-026-00747-y.

## Disclosure and competing interests statement
The authors declare no competing interests.

# Expanded View Figures

**Figure EV1. Effect of FBS on macrophage polarization.**

(**A**) Bone marrow cells from C57BL/6 mice were incubated with M-CSF (10 ng/ml) for seven days (M0 macrophages), then treated with LPS (10 ng/ml, 24 h) for M1-type polarization. The indicated mRNA expression levels were quantified by qPCR. Data were expressed as the ratio of 18 s RNA ($n = 4$–5). (**B**) Glycerol contents in FBS, dialyzed FBS (dFBS), and serum-free RPMI medium were measured using the high-sensitivity free glycerol fluorometric assay kit (Sigma-Aldrich, Cat# MAK270). (**C**) LPS-primed BMDMs were incubated in RPMI medium supplemented with FBS, dFBS, or no FBS for 24 h. The indicated mRNA expression levels were quantified by qPCR ($n = 6$). (**D**) M0-type BMDMs were stimulated with LPS (10 ng/mL) in medium containing FBS, dFBS, or serum-free medium for 6 h. The indicated mRNA expression levels were quantified by qPCR ($n = 6$). Data information: All data presented in Fig. EV1 are mean ± standard error of the mean (SEM). $N$ values indicate biological replicates. (**A**) Unpaired $t$-tests. (**C**) One-way ANOVA with Sidak's multiple comparison test. (**D**) One-way ANOVA with Dunnett's multiple comparison test. Exact P values are reported, except where the adjusted $P$ value was smaller than 0.0001, in which case it is reported as $P < 0.0001$. Source data are available online for this figure.

▶

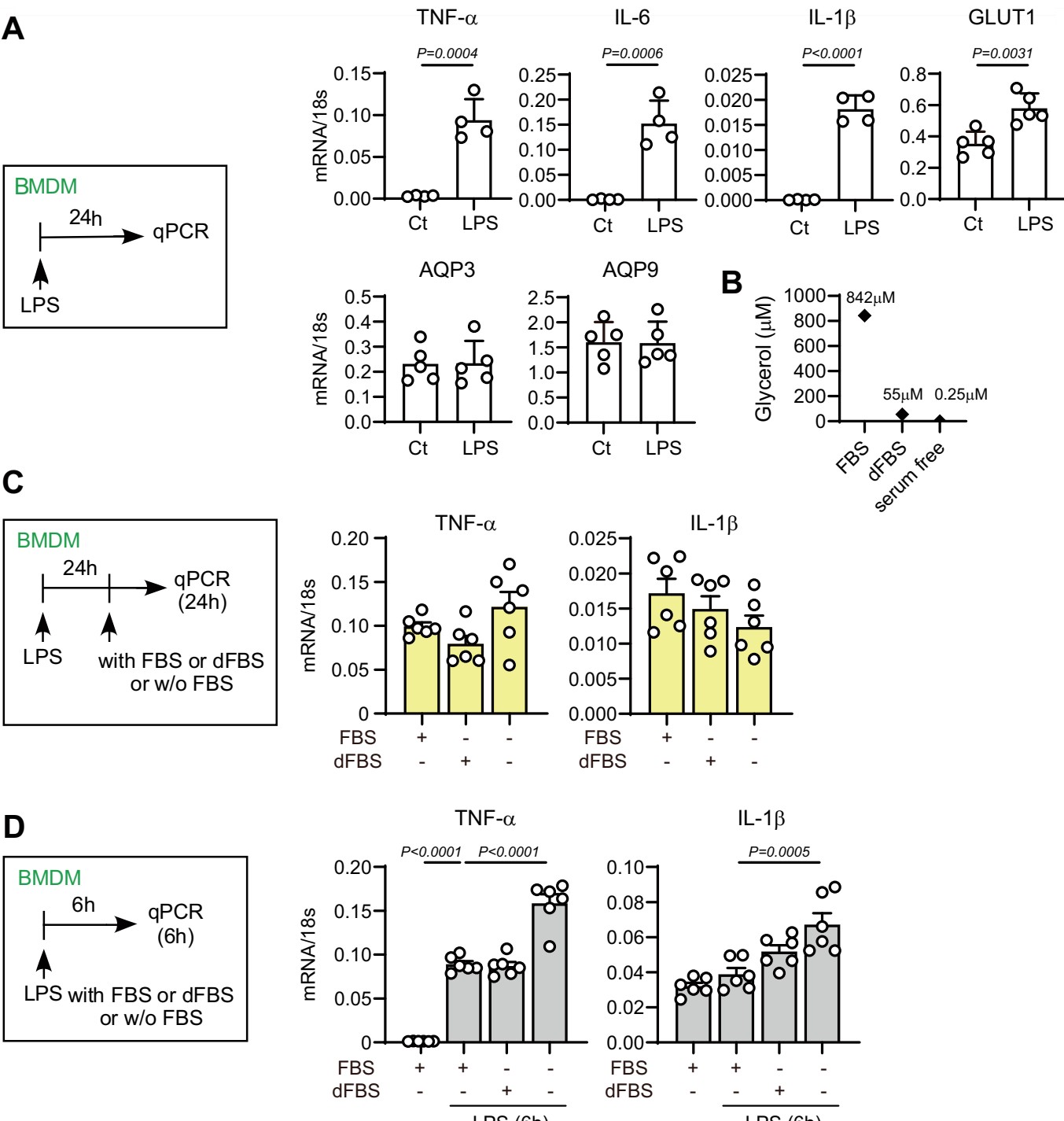

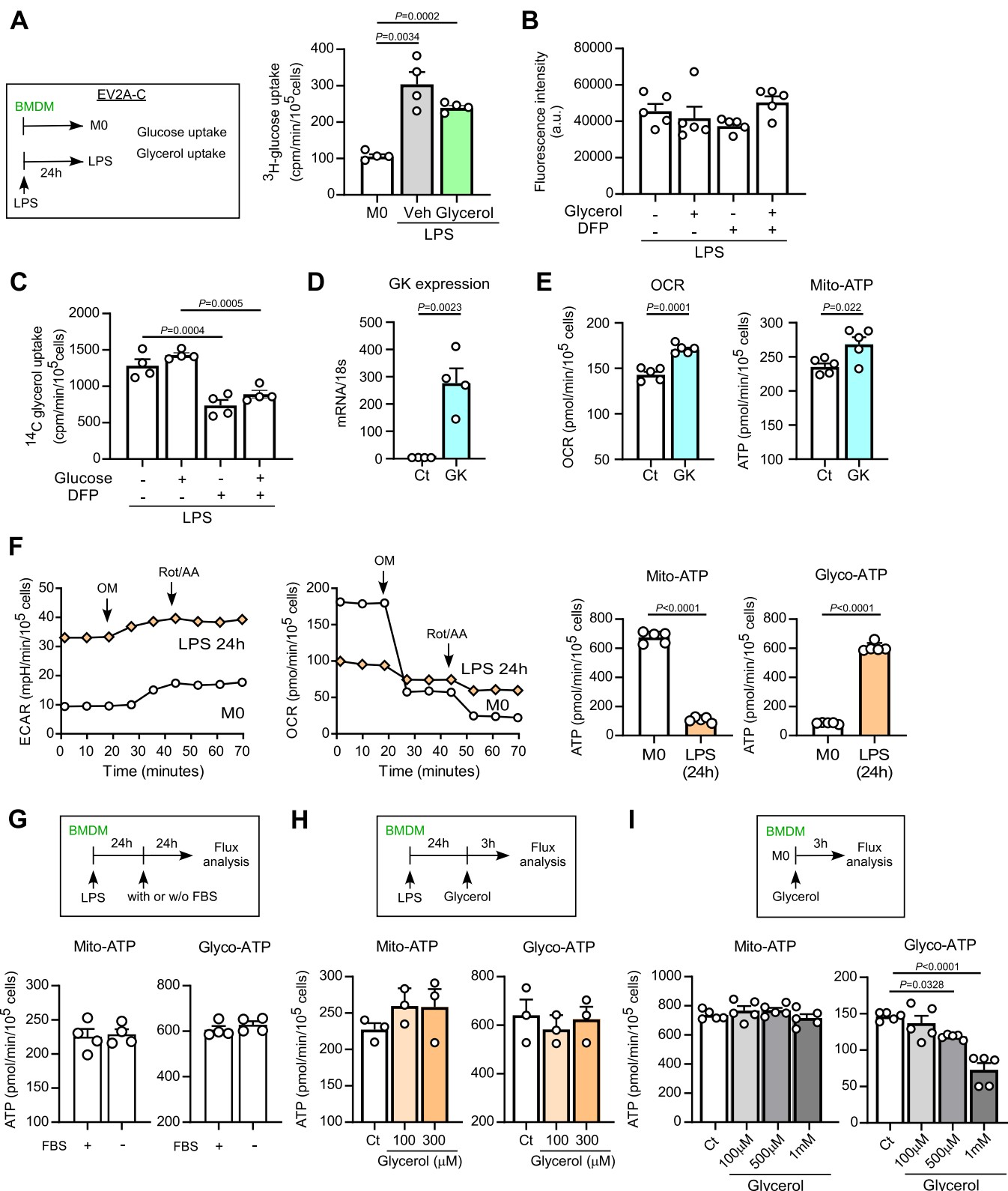

**Figure EV2. Glycerol does not affect glucose uptake.**

(A) [$^{3}$H]-glucose uptake for one hour in M0 and LPS-primed macrophages in the presence or absence of glycerol (1 mM). Cells were starved of glucose for at least one hour prior to the assay ($n = 4$). (B) LPS-primed macrophages were starved of glucose for at least one hour prior to the assay. Glucose uptake for 1 h was measured using a Glucose Uptake Assay Kit (DOJINDO, UP02) with glycerol (1 mM) and/or DFP00173 (1 mM) ($n = 5$). (C) [$^{14}$C]-glycerol uptake in LPS-primed macrophages. To assess whether glycerol uptake depends on intracellular glucose levels, cells were maintained in glucose-depleted medium for three hours prior to the assay, followed by incubation with [$^{14}$C]-glycerol for 3 min ($n = 4$). (D, E) BMDMs were overexpressing GK by adenovirus infection. (D) mRNA encoding for GK. Data were expressed as the ratio to 18 s RNA ($n = 4$). (E) Cellular OCR and mito-ATP production by ATP Rate Assay Kit in GK overexpressed LPS-primed macrophages in the presence of glycerol (1 mM) ($n = 5$). (F) BMDMs (M0) and LPS-primed BMDMs (M1) from wild-type mice were assessed using an ATP Rate Assay Kit with a Flux analyzer. (left) ECAR and OCR. (right) Mitochondria-derived and glycolysis-derived ATP production ($n = 5$). (G) LPS-primed BMDMs were incubated in medium supplemented with FBS or without FBS for 24 h. Mitochondria-derived and glycolysis-derived ATP production were analyzed using an ATP Rate Assay Kit with a Flux analyzer ($n = 4$). (H) LPS-primed BMDMs were incubated with glycerol (100 or 300 μM) in serum-free medium for 3 h. Mitochondria-derived and glycolysis-derived ATP production were analyzed using an ATP Rate Assay Kit with a Flux analyzer ($n = 3$). (I) M0-type BMDMs were incubated with glycerol (100, 500 μM or 1 mM) in serum-free medium for 3 h. Mitochondria-derived and glycolysis-derived ATP production were analyzed using an ATP Rate Assay Kit with a Flux analyzer ($n = 5$). Data information: All data presented in Fig. EV2 are mean ± standard error of the mean (SEM). N values indicate biological replicates. (A–C) One-way ANOVA with Sidak's multiple comparison test. (D–G) Unpaired t-tests. (H, I) One-way ANOVA with Dunnett's multiple comparison test. Exact P values are reported, except where the adjusted P value was smaller than 0.0001, in which case it is reported as $P < 0.0001$. Source data are available online for this figure.

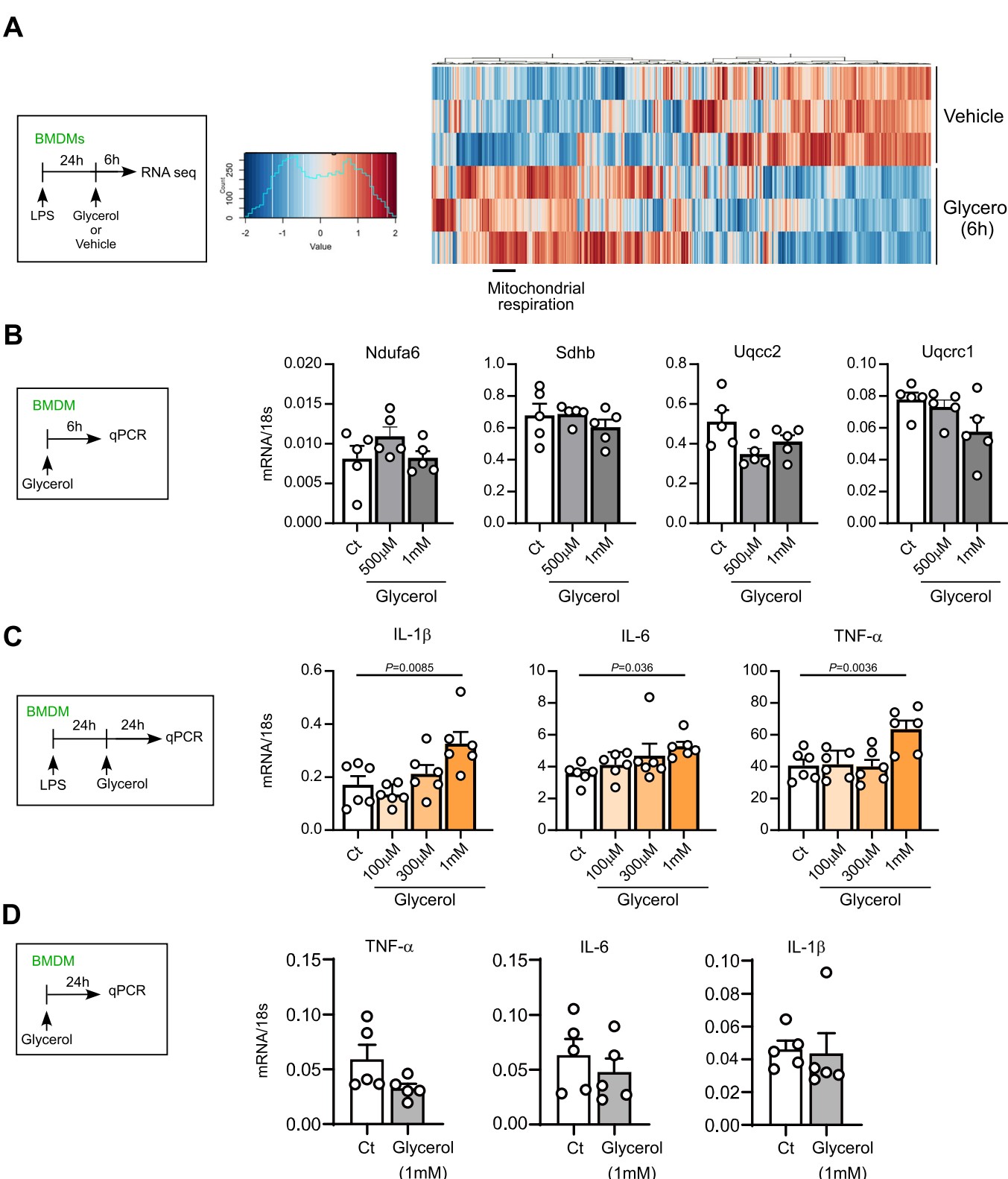

**Figure EV3. Glycerol alters gene expression in pro-inflammatory macrophages.**

(A) LPS-primed BMDMs were stimulated with glycerol (1 mM) for 6 h after glycerol starvation. RNA sequencing analysis was conducted, as shown in Fig. 3A. Differentially expressed genes ($Log_2$ [fold change] $\leq-2$ or $\geq2$) in control and glycerol-treated cells as summarized by heatmap. Red = upregulated; blue = downregulated ($n = 3$, biologically independent samples). (B) M0-type BMDMs were incubated with glycerol (500 µM or 1 mM) in serum-free medium for 6 h. The indicated mRNA expression levels were quantified by qPCR ($n = 5$). (C) LPS-primed BMDMs were incubated with glycerol (100, 300 µM or 1 mM) in serum free medium for 24 h. The indicated mRNA expression levels were quantified by qPCR ($n = 6$). (D) M0-type BMDMs were incubated with glycerol (1 mM) in serum free medium for 24 h. The indicated mRNA expression levels were quantified by qPCR ($n = 5$). Data information: All data presented in Fig. EV3 are mean ± standard error of the mean (SEM). *N* values indicate biological replicates. (B, C) One-way ANOVA with Dunnett's multiple comparison test. (D) Unpaired *t*-tests. Exact *P* values are reported. Source data are available online for this figure.

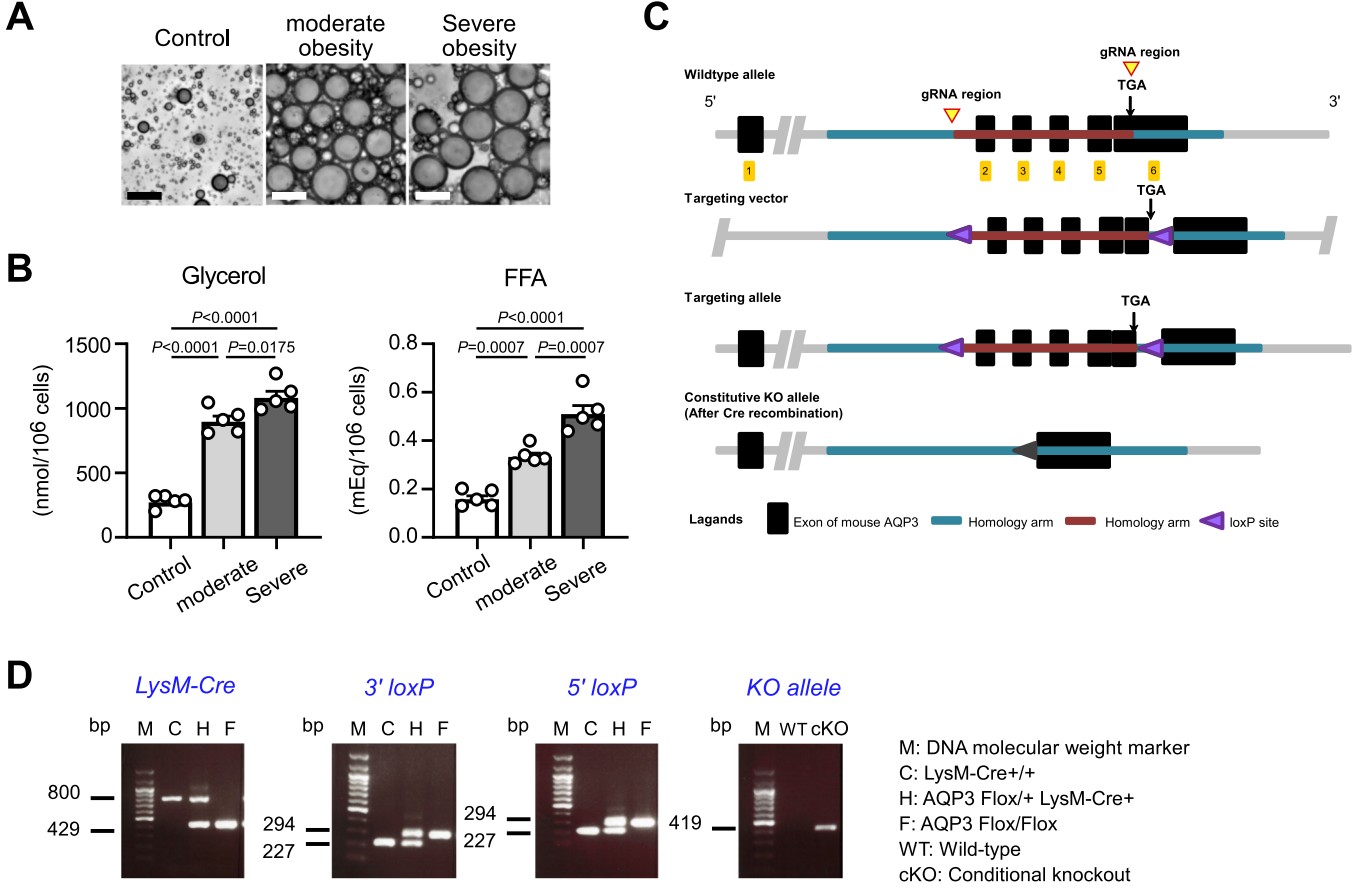

**Figure EV4. Generation of macrophage-specific AQP3 conditional knockout mice.**

(A) Images of adipocytes from control (NC) and HFD-induced obese mice (moderate: ~35-g weight, severe: ~50-g weight). (B) Glycerol and non-esterified fatty acid (FFA) content in the culture medium ($n = 5$). (C) Generation of AQP3$^{flox/flox}$ mice. Overview of the targeting strategy. (D) Representative detection of AQP3 deletion in AQP3$^{flox/flox}$ LysM-Cre$^+$ (AQP3 cKO) by PCR with genomic DNA. Data information: All data presented in Fig. EV4 are mean ± standard error of the mean (SEM). N values indicate biological replicates. (B) One-way ANOVA with Tukey's multiple comparisons test. Exact $P$ values are reported, except where the adjusted $P$ value was smaller than 0.0001, in which case it is reported as $P < 0.0001$. Source data are available online for this figure.

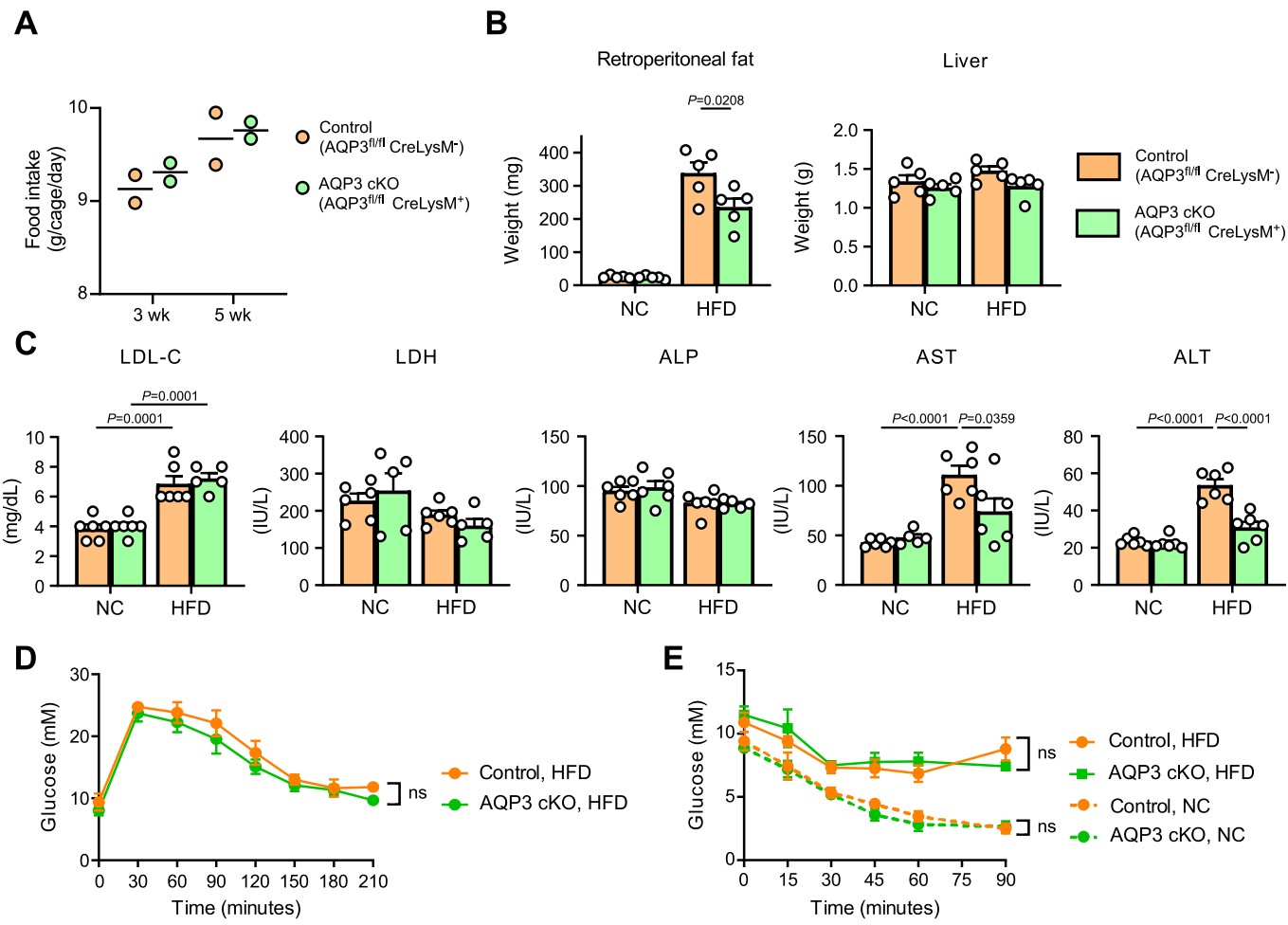

**Figure EV5. AQP3 deficiency in macrophages attenuates HFD-induced obesity.**

(A–E) AQP3 cKO and control mice were fed high fat diet (HFD) or normal chow (NC) for 6 weeks, same setting as shown in Fig. 5. (A) HFD chow consumption per cage for 24 h was measured at 3 and 5 weeks, demonstrating no difference between the two groups (two cages for each condition, three mice in each cage). (B) Weight of adipocyte tissues and liver ($n = 5$). (C) Low-density lipoprotein cholesterol (LDL-C), lactate dehydrogenase (LDH), alkaline phosphatase (ALP), aspartate aminotransferase (AST), and alanine aminotransferase (ALT) contents in serum ($n = 5$–6). (D) Glucose tolerance test in control and AQP3 cKO mice fed with HFD ($n = 3$). (ns not significant). (E) Insulin tolerance test in control and AQP3 cKO mice fed with NC or HFD ($n = 5$). (ns not significant). Data information: All data presented in Fig. EV5 are mean ± standard error of the mean (SEM). $N$ values indicate biological replicates. (B–D) Two-way ANOVA with Sidak's multiple comparisons test. (E) Two-way ANOVA with Tukey's multiple comparisons test. Exact $P$ values are reported, except where the adjusted $P$ value was smaller than 0.0001, in which case it is reported as $P < 0.0001$. Source data are available online for this figure.

