## [Peer Review File · EMBO Reports]

Glycerol enhances mitochondrial metabolism and inflammatory response in pro-inflammatory macrophages

Manami Tanaka, Takako Hishiki, Tomomi Matsuura, Masato Yasui, Shunsuke Chikuma, and Mariko Hara-Chikuma

Corresponding author(s): Mariko Hara-Chikuma (mariko.chikuma@keio.jp)

Review Timeline:

Submission Date:	27th Jun 25
Editorial Decision:	20th Aug 25
Revision Received:	27th Nov 25
Editorial Decision:	26th Jan 26
Revision Received:	28th Jan 26
Accepted:	2nd Mar 26

Editor: Deniz Senyilmaz Tiebe / Achim Breiling

Transaction Report:

Dear Dr. Hara-Chikuma,

Thank you for transferring your manuscript to EMBO Reports, which was now seen by three referees, whose reports are copied below. Please accept my apologies for the delay in getting back to you. It took longer than anticipated to receive the full set of referee reports given this busy time of the year.

Referees express interest in the proposed role of glycerol metabolism in regulation of mitochondrial function and proinflammatory responses in macrophages. However, they also raise important concerns that need to be addressed for publication here. In particular,

- Key experiments were performed at supraphysiological glycerol conditions (referee #3, standfirst and points 2 and 5)
- Discrepancy regarding the in vivo experiments need to be addressed as outlined by referee #3 (point 10).
- There are currently inconsistencies in the experimental conditions (referee #1, points 1, 2; referee #2, point 4).
- AQP3 cKO BMDMs and AQP3 cKO mice need better functional characterization (referee #1, points 3 and 4).
- Additional controls and experimental details need to be provided (all referees).

Please contact me if you have questions or comments regarding any of these points or the revision for further discussion (also by video chat).

In case you can satisfactorily address these concerns, we would like to invite you to revise your manuscript with the understanding that the referee concerns (as in their reports) must be fully addressed and their suggestions taken on board. Please address all referee concerns in a complete point-by-point response. Acceptance of the manuscript will depend on a positive outcome of a second round of review. It is EMBO reports policy to allow a single round of major experimental revision only and acceptance or rejection of the manuscript will therefore depend on the completeness of your responses included in the next, final version of the manuscript.

We realize that it is difficult to revise to a specific deadline. In the interest of protecting the conceptual advance provided by the work, we recommend a revision within 3 months. Please discuss the revision progress ahead of this time with me if you require more time to complete the revisions, or if you have questions or comments regarding the revision (also by video chat).

1. A data availability section providing access to data deposited in public databases is missing (where applicable).
2. Your manuscript contains statistics and error bars based on $n=2$. Please use scatter plots in these cases.

You can submit the revision either as a Scientific Report or as a Research Article. For Scientific Reports, the revised manuscript can contain up to 5 main figures and 5 Expanded View figures, and it should not exceed 27000 characters. If the revision leads to a manuscript with more than 5 main figures it will be published as a Research Article. In this case the Results and Discussion section should be separate. If a Scientific Report is submitted, these sections have to be combined. This will help to shorten the manuscript text by eliminating some redundancy that is inevitable when discussing the same experiments twice. In either case, all materials and methods should be included in the main manuscript file.

- Additional Tables/Datasets should be labeled and referred to as Table EV1, Dataset EV1, etc. Legends have to be provided in

a separate tab in case of .xls files. Alternatively, the legend can be supplied as a separate text file (README) and zipped together with the Table/Dataset file.

4) a .docx formatted letter INCLUDING the reviewers' reports and your detailed point-by-point responses to their comments. As part of the EMBO publication's Transparent Editorial Process, EMBO reports publishes online a Review Process File (RPF) to accompany accepted manuscripts. This File will be published in conjunction with your paper and will include the referee reports, your point-by-point response and all pertinent correspondence relating to the manuscript.

<https://www.embopress.org/page/journal/14693178/authorguide#transparentprocess>

5) a complete author checklist, which you can download from our author guidelines

<https://www.embopress.org/page/journal/14693178/authorguide>. Please insert information in the checklist that is also reflected in the manuscript. The completed author checklist will also be part of the RPF.

6) Please note that all corresponding authors are required to supply an ORCID ID for their name upon submission of a revised manuscript (<<https://orcid.org/>>). Please find instructions on how to link your ORCID ID to your account in our manuscript tracking system in our Author guidelines

<<https://www.embopress.org/page/journal/14693178/authorguide#authorshippinguidelines>>

7) Before submitting your revision, primary datasets produced in this study need to be deposited in an appropriate public database (see <https://www.embopress.org/page/journal/14693178/authorguide#datadeposition>). Please remember to provide a reviewer password if the datasets are not yet public. The accession numbers and database should be listed in a formal "Data Availability" section placed after Materials & Method (see also

<https://www.embopress.org/page/journal/14693178/authorguide#datadeposition>). Please note that the Data Availability Section is restricted to new primary data that are part of this study. * Note - All links should resolve to a page where the data can be accessed. *

Additional information on source data and instruction on how to label the files are available:

<https://www.embopress.org/page/journal/14693178/authorguide#sourcedata>

9) Our journal encourages inclusion of *data citations in the reference list* to directly cite datasets that were re-used and obtained from public databases. Data citations in the article text are distinct from normal bibliographical citations and should directly link to the database records from which the data can be accessed. In the main text, data citations are formatted as follows: "Data ref: Smith et al, 2001" or "Data ref: NCBI Sequence Read Archive PRJNA342805, 2017". In the Reference list, data citations must be labeled with "[DATASET]". A data reference must provide the database name, accession number/identifiers and a resolvable link to the landing page from which the data can be accessed at the end of the reference. Further instructions are available at <http://www.embopress.org/page/journal/14693178/authorguide#referencesformat>

10) Regarding data quantification (see Figure Legends:

<https://www.embopress.org/page/journal/14693178/authorguide#figureformat>)

- the name of the statistical test used to generate error bars and P values,

- the number (n) of independent experiments (please specify technical or biological replicates) underlying each data point,

- the nature of the bars and error bars (s.d., s.e.m.),

- If the data are obtained from n Program fragment delivered error ``Can't locate object method "less" via package "than" (perhaps you forgot to load "than"?) at //ejpvfs23/sites23b/embor_www/letters/embor_decision_revise_and_review.txt line 56.' 2, use scatter blots showing the individual data points.

12) Please also note our reference format:

13) All Materials and Methods need to be described in the main text using our 'Structured Methods' format, which is required for all research articles. According to this format, the Methods section includes a Reagents and Tools Table (listing key reagents, experimental models, software and relevant equipment and including their sources and relevant identifiers) followed by a Methods and Protocols section describing the methods using a step-by-step protocol format. The aim is to facilitate adoption of the methodologies across labs. More information on how to adhere to this format as well as a downloadable template (.docx) for the Reagents and Tools Table can be found in our author guidelines:

I look forward to seeing a revised version of your manuscript when it is ready. Please let me know if you have questions or comments regarding the revision.

Kind regards,

Deniz Senyilmaz Tiebe

Deniz Senyilmaz Tiebe, PhD
Senior Scientific Editor
EMBO Reports

Referee #1:

This work examines the role of extracellular glycerol in macrophage mitochondrial metabolism and inflammatory gene expression mediated by the glycerol channel AQP3. The authors show that BMDMs readily take up glycerol, which could be partially blocked by AQP3 inhibition. Metabolite flux analysis indicates that glycerol is utilized to produce metabolites in glycolysis and G3P shuttle pathways. The authors went on to demonstrate that glycerol induces mitochondrial respiration that is accompanied by increased expression of OXPHOS genes. In addition, glycerol-derived acetyl-CoA enhances histone acetylation of inflammatory gene promoters during LPS stimulation, leading to an increase in the expression of inflammatory genes, such as IL-6 and IL-1b. Lastly, macrophage-specific AQP3 knockout (AQP3 cKO) mice are shown to gain less weight and have lowered circulating FFA and insulin levels on a HFD compared to WT mice. These improvements in metabolic parameters could be partly explained by reduced meta-inflammation in adipose tissue resident macrophages.

The study provides new insight into how glycerol may contribute to macrophage inflammatory response. There are, however, a few issues that require clarification.

Specific comments:

1. The authors were not quite consistent in how they conducted various experiments. For example, while the effect of glycerol on mitochondrial respiration and RNAseq studies (Fig. 2/3) were examined 24 hours after LPS treatment, the histone acetylation (ChIP-pcr) and proinflammatory gene expression (Fig. 4) were studied with LPS and glycerol cotreatment. The glycerol uptake and metabolomics analyses were conducted without LPS. To support their conclusion, the authors need to repeat experiments in dialyzed FBS (to remove glycerol) LPS glycerol during a time course from 2-24 hours.

2. LPS treatment is known to suppress mitochondrial oxidative metabolism. The reference (Langston et al 2019 Nat Immunol 20:1186) authors cited throughout the manuscript suggested that the glycerol phosphate shuttle is being activated immediately after LPS stimulation to drive mitochondrial OXPHOS that produces acetyl-CoA for histone acetylation of pro-inflammatory gene promoters. Since the current study is built upon findings from the cited paper, the authors should also conduct a time course study for mitochondrial respiration, metabolomics and metabolic flux analyses.

3. The authors need to conduct additional function studies to validate the AQP3 cKO. Key experiments in Fig. 1-4 using the

chemical inhibitor should be repeated in AQP3 cKO BMDMs.

4. The differences in metabolic parameters and inflammatory gene expression in Fig. 5 could be secondary to the weight difference between control and AQP3 cKO. What causes the reduced weight gain in AQP3 cKO mice? Do they eat less, move more or have higher metabolic rates? Do these mice have activated brown/beige adipocytes and if so, why? Additional metabolic studies are needed to fully characterize the metabolic phenotype of AQP3 cKO mice.

Minor points:

1. For RNAseq, the authors need to provide the numbers and list of up-regulated and down-regulated genes, the set(s) of genes used for KEGG analysis and the gene list for each KEGG category. Without this information, it is difficult to determine whether OXPHOS is the most relevant pathways altered.

Referee #2:

The manuscript by Tanaka et al. is an interesting study looking at the effect of extracellular glycerol and its oxidation on macrophage activation. Changes in mitochondrial metabolism are tightly associated with macrophage activation state, but the role of glycerol oxidation in macrophage activation is unknown. The findings are important given macrophage polarization is now thought to play a key determinative role in the progression of diet-induced obesity and maintenance of overall adipose tissue health. Moreover, on a molecular level, how lipid oxidation interfaces with the innate immune system and, specifically macrophage polarization, has been a subject of intense debate for well over a decade.

As such, this manuscript showing glycerol uptake and oxidation can play a role in macrophage metabolism is a useful addition to the field. In particular, the *in vivo* findings with a macrophage specific knockout of AQP3 attenuating the effects of diet-induced obesity establish the relevance of many of the *in vitro* results. There are a few areas, however, that the authors could address to strengthen the findings and improve the overall manuscript.

MAJOR COMMENTS

(1) There should be some additional controls done to show that the "glycerol deprivation," which appears to be serum starvation to remove glycerol, does not adversely harm the macrophages. Even though the authors show there is an inflammatory response from LPS, is it similar to the response in macrophages without serum starvation? Are there any other supplements (i.e. B27) or delipidated medium that would be more appropriate than serum starvation that would essentially remove all fatty acids and other growth factors and not just glycerol? This serum starvation might also change the concentration-response to LPS, as these cells seem to have a strong metabolic response to 10 ng/mL LPS whereas many others require higher concentrations. The authors should examine how serum starvation affects both the LPS response and the subsequent metabolic effects compared to a standard protocol where FBS is not removed from the medium.

(2) The authors should normalize all Seahorse data to cell number since LPS causes growth arrest of macrophages (usually reported as O₂, mpH, or ATP/min/1,000 cells). The authors also do not show whether or not glycerol has an effect on the control (no LPS) macrophages (M0s) in Figure EV3, which would be important to clarify if any effects from glycerol only occur in the LPS-treated condition.

(3) Figure 3 nicely shows regulation of relevant ETC genes by glycerol, and the specific regulation of complex III by glycerol and its inhibitors. The authors should demonstrate whether this regulation is specific to LPS treated macrophages and whether it occurs in control, M0 untreated macrophages. Furthermore, the result is striking because the G-3-P shuttle bypasses complex I and feeds electrons directly into the quinone pool. The authors should comment on how the specific regulation of complex III lines up well with the enzymes involved in G-3-P oxidation. Moreover, it would be logical since LPS activation causes nitric oxide-linked inhibition of complex I, thereby increasing reliance on glycerol oxidation upon LPS since the G-3-P shuttle feeds electrons straight into complex III via G3PDH.

(4) One note about the manuscript is that it is sometimes difficult to follow which time post-LPS treatment the authors are measuring and why. This is quite important given the temporal nature of the metabolic rewiring and histone acetylation (Ball et al. *EMBO Rep* (2025) 26: 982 - 1002; Lauterbach et al. *Immunity* 2019). In addition to clearly stating at what time each experiment was done and why, the authors should also examine the time dependence of the glycerol-induced increase of the pro-inflammatory response. Does this only happen at 24-48 hours, or what is the lag time between increased oxidation (earlier than 3 hours) and enhanced pro-inflammatory gene expression (I think measured at 24 hr) and cytokine release (48 hr.)

(5) The introduction is a bit outdated with regard to the field's understanding of mitochondrial energetics in macrophage polarization and may have some logical inconsistencies. If LPS shifts metabolism away from OxPhos, then why would increasing OxPhos enhance the pro-inflammatory response? A fuller picture including work from Ran et al [*Nature Metabolism* volume 5, pages804-820 (2023); which stands in contrast to the studies of Lauterbach], and work from Ball et al (cited in #4 and suggests that many metabolic phenotypes can accompany pro-inflammatory macrophage activation) would be helpful. The introduction should cite these works and give a more full picture of the field, and adjust the introduction to be more logically

consistent with the findings.

MINOR

- (a) Given that the M+1 isotopologue for acetyl CoA is quite high, the authors should mention in the methods how they correct for natural heavy isotopes in the methods.
- (b) The authors should cite their method for calculating ATP production rates from the XF Analyzer. If using the company's kit, this is Desousa et al. EMBO Reports 2023 Oct 9;24(10):e56380.

Referee #3:

This manuscript is an investigation into the AQP3-mediated import of glycerol and its downstream metabolism in inflammatory macrophage metabolism and phenotype. It is a very interesting, timely and potentially physiologically relevant study. However, there are aspects of the study - in particular the non-physiologically high glycerol concentrations used in vitro, and that make the physiological relevance a little distant. Additionally, the metabolic results of the in vivo work are very exciting, somewhat unexpected, and not discussed in enough detail by the authors - given this unexpected nature (outlined below).

Major comments

1. Could the authors clarify what is meant in the Figure Legend for EV2 'Some cells were glucose-depleted overnight prior to the assay'. It isn't clear which assay, or whether this might produce some intra-assay variation that isn't otherwise obvious from the axis labels.
2. The authors state that on analysis, their FBS contains 1 mM glycerol. This is incredibly high and non-physiological for mammalian species - indeed in humans it is around 0.1 mM (Stinshoff et al. Clinical Chemistry 1977; Hagstrom-Toft et al. Endocrinology and Metabolism, 1997)). It is therefore confusing as to how this concentration was arrived at - whether bad batches of FBS were examined, or analytical error in measurement. Both would have implications for the manuscript as a whole. If the physiological concentration of glycerol of 0.1 mM is correct in the sera, then the relevance of this study is under very specific conditions (given the results in Figure 1A) - which would be likely observed in lipolytic adipose tissue.
3. In the initial experiment with a dose response for glycerol, the authors incubate the macrophages with the 14C-glycerol for 3 minutes. It isn't clear if the authors have data showing that this time is sufficient to reach steady-state concentrations in the cells, or whether if longer times were used, the profile of their experiment might change. A similar point is true for the 13C-glycerol experiment - the metabolites they report as being labelled - do they know that they have reached steady state for these as well? It might well rely on the kinetics of glycerol kinase, and how readily the cells activate the glycerol for further metabolism in glycolysis
4. When the authors examine the response of glycolysis and mitochondrial ATP generation mechanisms in response to LPS and then glycerol, they suggest that their results show that 'glycerol was able to restore mitochondrial ATP production...following LPS-induced suppression.' The suppression by LPS was recorded by the authors as around 3-fold reduction in OCR, accompanied by a 3-fold increase in EACR (EV3A). The change from this basal rate after glycerol was found to be ~33% increase in OCR (and mitoATP in 2A-C) and a further ~15% in EACR. The authors might consider altering their language to more accurately represent the increase in OCR by glycerol, which would need to be ~300% increase to restore mitochondrial ATP production.
5. In the glycerol-starved control conditions, the authors were below their detection limit (although no data are shown to support this), which is 1 μ M or less. In Figure 2E-G they suggest that their results show that physiological glycerol levels are too low for inflammatory macrophages to utilise glycerol for energy production. However, for this experiment they did not test physiological levels of glycerol, but less than 1 μ M - 100-fold lower than the serum concentration. In order to make this conclusion, they need to test the effect of inhibitors on physiological levels - i.e. around 100 μ M.
6. In the GK experiment (EV2D-E) I could not find the extracellular concentrations of glycerol that were used in this assay - could this be corrected.
7. There is no section on analysis of the data arising from the metabolomic studies - this must be corrected
8. As a result of the successful inhibition of histone acetylation by the GPD2 inhibitor, the authors point out that this suggests that the glycerol 3-phosphate shuttle is therefore implicated in this process. It is unclear how this could mechanistically occur, given that it donates electrons directly to Ub. Could the authors suggest in the discussion how this might occur?
9. The authors cite two papers in support of elevated plasma glycerol levels in obese individuals - van der Merwe 1999 and 2001. The former describes glycerol in lean individuals, and the latter in lean, obese and diabetic women only. Not only that, the measurements taken show an increase in plasma glycerol from 55 μ M to 80 μ M between lean and obese individuals. This cannot be used to justify the hypothesis that the in vitro results using 1 mM glycerol could be recapitulated in obese animals. Could the authors profile glycerol in the plasma of the obese mice to show the contribution of the significant increase in lipolysis they show in EV4B to systemic concentrations?
10. The use of the AQP3 mouse is based around the inhibition of around 40% of glycerol uptake by DFP by macrophages in vitro and 50% in vivo in the AQP3 KO. Given this, it is surprising that the significant changes in fat content and metabolites were observed in the serum - indeed suggesting that loss of 50% of the glycerol transport function only in myeloid cells results in unexpectedly large metabolic defects - particularly decreased serum glycerol. This appears at odds with the hypothesis that macrophage import of glycerol is a key physiological feature, particularly with HFD, and this is not explained by the authors in the discussion. This is a critical discrepancy that should be resolved experimentally or explained in the discussion.

Minor comments

11. Contrary to the first line of the introduction, the metabolic pathway of glycerol is indeed well-described. If the authors wanted to say something different - perhaps that any specific metabolic pathways used in macrophages from glycerol (which is indeed not yet documented) - they should say this more specifically.
12. Y-axis in EV3A should read pmol/min
13. Figure 3F x axis has 'TG' instead of thio in the vehicle condition

Response to Editors and Referees

We thank the editor and Referees for their thoughtful comments on our manuscript. We greatly appreciate the effort and time taken to provide us with a number of constructive comments, which have helped us significantly improve the quality of our article. We are particularly grateful that the editor carefully integrated the overlapping concerns raised by the three Referees, which made us aware of important shortcomings in the original version. We realized that, although the primary aim of this study was to investigate the effects of high concentrations of glycerol on pro-inflammatory macrophages, the inclusion of data on the effects of glycerol on M0 macrophages inadvertently shifted the focus away from this central objective. In the revised version, we therefore performed additional experiments and now focus all main figures exclusively on the impact of glycerol on pro-inflammatory (LPS-primed) macrophages, in line with the original scope of the study. To clarify these changes, we provide below a summary table listing the experiments presented in the original and revised versions of the manuscript, with newly added data highlighted in red. For the convenience of readers, we have also included this table in the revised manuscript as an Appendix Table S1. We have carefully addressed all concerns raised by the three Referees in the revised manuscript and in the accompanying point-by-point response. (Author's responses are shown in blue).

Table for referee with unpublished data has been removed upon request by the authors.

Referee #1:

This work examines the role of extracellular glycerol in macrophage mitochondrial metabolism and inflammatory gene expression mediated by the glycerol channel AQP3. The authors show that BMDMs readily take up glycerol, which could be partially blocked by AQP3 inhibition. Metabolite flux analysis indicates that glycerol is utilized to produce metabolites in glycolysis and G3P shuttle pathways. The authors went on to demonstrate that glycerol induces mitochondrial respiration that is accompanied by increased expression of OXPHOS genes. In addition, glycerol-derived acetyl-CoA enhances histone acetylation of inflammatory gene promoters during LPS stimulation, leading to an increase in the expression of inflammatory genes, such as IL-6 and IL-1b. Lastly, macrophage-specific AQP3 knockout (AQP3 cKO) mice are shown to gain less weight and have lowered circulating FFA and insulin levels on a HFD compared to WT mice. These improvements in metabolic parameters could be partly explained by reduced meta-inflammation in adipose tissue resident macrophages. The study provides new insight into how glycerol may contribute to macrophage inflammatory response. There are, however, a few issues that require clarification.

Specific comments:

1. The authors were not quite consistent in how they conducted various experiments. For example, while the effect of glycerol on mitochondrial respiration and RNAseq studies (Fig. 2/3) were examined 24 hours after LPS treatment, the histone acetylation (ChIP-pcr) and proinflammatory gene expression (Fig. 4) were studied with LPS and glycerol cotreatment. The glycerol uptake and metabolomics analyses were conducted without LPS. To support their conclusion, the authors need to repeat experiments in dialyzed FBS (to remove glycerol) \pm LPS \pm glycerol during a time course from 2-24 hours.

RESPONSE:

We thank the reviewer for pointing out the inconsistency in the experimental conditions across Figures 1–4. We fully agree that the mixture of M0 and M1 macrophages in the original version obscured the central message of the study. The primary goal of our work is to determine how glycerol affects inflammatory macrophages, namely LPS-primed M1 macrophages. However, as the reviewer correctly noted, we had unintentionally included experiments examining the effects of glycerol during the M0-to-M1 priming phase (e.g., ChIP-PCR and proinflammatory gene induction), while other experiments were performed in fully primed M1 macrophages. We now recognize that this inconsistency caused confusion and detracted from the main focus of the manuscript.

In the revised version, results conducted on M0 macrophages have been either removed or transferred to the Extended Figure. Instead, we repeated the key experiments—histone acetylation

(ChIP–PCR) and inflammatory cytokine gene expression—using LPS-primed M1 macrophages. The new results (Fig. 4A, 4B) demonstrate that glycerol enhances inflammatory gene expression even in fully primed M1 macrophages and that inhibition of the G3P shuttle suppresses this effect. All main figures in the revised manuscript now focus on LPS-primed inflammatory macrophages, whereas results from M0 macrophages are shown only in the Expanded View figures. Schematic diagrams were added to each figure to clarify the experimental design.

As suggested by the reviewer, we also performed additional experiments using dialyzed FBS to deplete glycerol. Dialysis reduced FBS glycerol levels by >90% (Figure EV1B). When LPS-primed macrophages were incubated for 24 h in medium containing dialyzed FBS, the expression levels of the M1 inflammatory markers TNF and IL-1 β were not altered; similarly, incubation in FBS-free medium had no effect on these markers (Figure EV1C). Using dialyzed FBS would have been an attractive option; however, because molecules smaller than 10 kDa are removed, we considered that this would make data interpretation difficult. We therefore standardized our assay conditions by fully priming macrophages with LPS for 24 h and subsequently starving them in FBS-free medium for at least 1 h before each assay.

Together, these revisions unify all key data under a single experimental framework—LPS-primed inflammatory macrophages—and directly address the reviewer’s concerns about experimental consistency.

2. LPS treatment is known to suppress mitochondrial oxidative metabolism. The reference (Langston et al 2019 Nat Immunol 20:1186) authors cited throughout the manuscript suggested that the glycerol phosphate shuttle is being activated immediately after LPS stimulation to drive mitochondrial OXPHOS that produces acetyl-CoA for histone acetylation of pro-inflammatory gene promoters. Since the current study is built upon findings from the cited paper, the authors should also conduct a time course study for mitochondrial respiration, metabolomics and metabolic flux analyses.

RESPONSE:

We thank the reviewer for this insightful comment. In the original version, we separately examined the effects of glycerol on LPS-primed macrophages and on macrophages during the LPS-driven transition from M0 to M1 and originally presented the impact of glycerol on each of these phases with different time course.

As we explain in our response to comment 1, however, the main objective of the present study is to investigate how elevated glycerol affects already established pro-inflammatory (LPS-primed) macrophages, rather than the early priming process itself. In the revised manuscript, we therefore focused our analyses on LPS-primed macrophages that were stimulated with LPS for 24 h and then exposed to glycerol. In new experiments under these conditions, a 3-h glycerol incubation

increased histone acetylation in a G3P-shuttle–dependent manner (New Fig. 4A). Moreover, metabolomics, metabolic flux analysis, and qPCR were all include data obtained at the same time point (24-h LPS priming followed by 3-h glycerol treatment), and consistently supported the conclusion that glycerol augments inflammatory metabolism through the G3P shuttle.

We fully agree that a detailed time-course analysis from 2 to 24 h after LPS stimulation, as performed by Langston et al., would be highly informative for understanding the dynamics of the priming phase. However, because our revised study is now specifically focused on the late phase in fully LPS-primed macrophages and on the impact of high glycerol in this context, we did not extend the current work to a full time-course.

We have substantially revised the manuscript, particularly the description of Figure 4, as well as the Introduction and Discussion sections. These revisions clarify the overall objectives of the study and present the principal experimental findings and their interpretation in a clearer and more straightforward manner.

3. The authors need to conduct additional function studies to validate the AQP3 cKO. Key experiments in Fig. 1-4 using the chemical inhibitor should be repeated in AQP3 cKO BMDMs.

RESPONSE:

Thank you for this important comment. To validate the AQP3 cKO macrophages, we first examined whether AQP3 deficiency affects LPS-induced M1 polarization. In AQP3 cKO BMDMs, LPS stimulation markedly reduced the induction of canonical M1 markers, including Il1b, Il6, Tnf, Nos2, and others, indicating that AQP3-deficient macrophages exhibit impaired priming and incomplete differentiation into the M1 state (New Figure 4F).

We next performed extracellular flux analysis. WT macrophages displayed robust increases in ECAR and glycolytic ATP production after 24 h of LPS stimulation, consistent with metabolic reprogramming toward the M1 phenotype. In contrast, AQP3 cKO macrophages showed blunted increases in both parameters, further supporting that the metabolic program required for full M1 differentiation is substantially impaired in these cells (New Figure 4G).

Because most experiments in Figures 1–4 were specifically designed to evaluate the effect of glycerol in fully primed M1 macrophages (LPS 24 h), and because AQP3 cKO macrophages do not reach a comparable M1 state, it was not technically or biologically appropriate to repeat these assays in the cKO cells, as the baseline phenotype differs fundamentally from WT.

However, these new findings provide an additional layer of insight: AQP3 appears to be required not only for glycerol-dependent metabolic responses but also for optimal M1 polarization itself. We added these new data and incorporated a discussion on the possibility that glycerol–AQP3–GPS signaling may contribute to the establishment of the M1 phenotype.

4. The differences in metabolic parameters and inflammatory gene expression in Fig. 5 could be secondary to the weight difference between control and AQP3 cKO. What causes the reduced weight gain in AQP3 cKO mice? Do they eat less, move more or have higher metabolic rates? Do these mice have activated brown/beige adipocytes and if so, why? Additional metabolic studies are needed to fully characterize the metabolic phenotype of AQP3 cKO mice.

RESPONSE:

We appreciate the reviewer's thoughtful comments regarding the metabolic phenotype of AQP3 cKO mice and the potential impact of body-weight differences on the results shown in Fig. 5. To address these points, we performed additional analyses.

First, to determine whether reduced weight gain in AQP3 cKO mice is due to decreased food intake, we monitored HFD consumption and found no significant difference between control and AQP3 cKO mice (new Fig. EV5A). We also carried out glucose and insulin tolerance tests, which revealed comparable glucose tolerance and insulin sensitivity between the two genotypes (new Fig. EV5D, E). These data suggest that the reduced weight gain in AQP3 cKO mice is not explained by changes in caloric intake or by major alterations in systemic glucose homeostasis under our experimental conditions.

To further investigate the mechanism, we re-analyzed gene expression in epididymal white adipose tissue. In control mice, HFD feeding markedly increased the expression of M1 macrophage markers (Tnf, Il1b, Nos2, Ccl2, Ccr7), whereas this induction was significantly attenuated in AQP3 cKO adipose tissue. These results indicate that obesity-associated inflammatory activation of adipose tissue macrophages is clearly reduced in AQP3 cKO mice and support the interpretation that the attenuation of weight gain is at least partly secondary to diminished adipose tissue inflammation rather than to primary changes in energy intake or glucose handling.

As the reviewer suggests, comprehensive metabolic phenotyping, including indirect calorimetry and analysis of brown/beige adipose tissue, will be important future work to fully characterize the metabolic phenotype of AQP3 cKO mice. We now explicitly acknowledge this limitation in the Discussion and have toned down our statements regarding systemic metabolic effects, and emphasize that our current data most strongly support a role for AQP3 in promoting HFD-induced adipose tissue inflammation and that the leaner phenotype of AQP3 cKO mice is likely a secondary consequence of this reduced inflammatory state (Page 15, line 31-).

Minor points:

1. For RNAseq, the authors need to provide the numbers and list of up-regulated and down-regulated genes, the set(s) of genes used for KEGG analysis and the gene list for each KEGG

category. Without this information, it is difficult to determine whether OXPHOS is the most relevant pathways altered.

RESPONSE:

We have now provided the numbers of up- and down-regulated genes in the revised results section (page 10, line 4-) , the lists of upregulated and downregulated genes (Appendix Table S2, S3), the gene list for mitochondrial metabolic pathways (Appendix Table S4), and the set of genes used for KEGG analysis (Source data set Fig. 3A).

Raw RNA-seq data have been deposited in the DDBJ Sequence Read Archive under BioProject accession number PRJDB39611.

Referee #2:

The manuscript by Tanaka et al. is an interesting study looking at the effect of extracellular glycerol and its oxidation on macrophage activation. Changes in mitochondrial metabolism are tightly associated with macrophage activation state, but the role of glycerol oxidation in macrophage activation is unknown. The findings are important given macrophage polarization is now thought to play a key determinative role in the progression of diet-induced obesity and maintenance of overall adipose tissue health. Moreover, on a molecular level, how lipid oxidation interfaces with the innate immune system and, specifically macrophage polarization, has been a subject of intense debate for well over a decade.

As such, this manuscript showing glycerol uptake and oxidation can play a role in macrophage metabolism is a useful addition to the field. In particular, the *in vivo* findings with a macrophage specific knockout of AQP3 attenuating the effects of diet-induced obesity establish the relevance of many of the *in vitro* results. There are a few areas, however, that the authors could address to strengthen the findings and improve the overall manuscript.

MAJOR COMMENTS

(1) There should be some additional controls done to show that the "glycerol deprivation," which appears to be serum starvation to remove glycerol, does not adversely harm the macrophages. Even though the authors show there is an inflammatory response from LPS, is it similar to the response in macrophages without serum starvation? Are there any other supplements (i.e. B27) or delipidated medium that would be more appropriate than serum starvation that would essentially remove all fatty acids and other growth factors and not just glycerol? This serum starvation might also change the concentration-response to LPS, as these cells seem to have a strong metabolic response to 10 ng/mL LPS whereas many others require higher concentrations. The authors should examine how serum starvation affects both the LPS response and the subsequent metabolic effects compared to a standard protocol where FBS is not removed from the medium.

RESPONSE:

We thank the reviewer for this important comment. We agree with these comments and have now performed additional controls to address this concern.

In LPS-primed macrophages, 24 h incubation with the medium in the presence or absence of FBS did not alter the expression of inflammatory markers such as TNF- α and IL-1 β (new Fig. EV1C), nor did it affect ATP production measured by extracellular flux analysis (new Fig. EV2G).

We also explored alternative approaches to remove glycerol. Delipidated serum did not efficiently reduce glycerol, whereas dialyzed FBS lowered glycerol by ~90% (new Fig. EV1B). Dialysis FBS also did not affect the mRNA expressions of TNF- α and IL-1 β in LPS-primed macrophages (new Fig. EV1C). Because serum starvation did not markedly affect inflammatory or metabolic

parameters in LPS-primed M1-type macrophages, all subsequent experiments were performed by first differentiating M0 cells into LPS-primed macrophages for 24 h in complete medium and then switching them to FBS-free medium for 1-3 hours before adding glycerol.

As suggested by the reviewer, we examined the effect of glycerol on M0 macrophages and found that FBS free medium enhanced LPS-induced TNF- α and IL-1 β expression (new Fig. EV1D), indicating increased sensitivity to LPS stimulation in M0 macrophages under serum-free conditions.

In response to your comment #4, we realized that the original version mixed data obtained under different conditions (M0 macrophages during LPS priming and LPS-primed pro-inflammatory macrophages), which obscured our main objective of examining the effects of high concentrations of glycerol on pro-inflammatory macrophages.

In the revised manuscript, we therefore restricted all main figures to experiments performed in LPS-primed pro-inflammatory macrophages, and moved the data on glycerol effects during the M0-to-M1 priming phase to the Expanded View figures or removed them when they were not essential. The additional experiments performed in response to this comment have been included in the Expanded View figures (EV1B, 1C, 1D, 2G), and we now clearly state that LPS-primed macrophages were incubated in FBS-free medium for 1–3 h prior to the assay to deplete extracellular glycerol. This unification and clearer description of the experimental conditions could make it easier to follow the logic of the study and to interpret the effects of glycerol on inflammatory macrophages.

(2) The authors should normalize all Seahorse data to cell number since LPS causes growth arrest of macrophages (usually reported as O₂, mpH, or ATP/min/1,000 cells). The authors also do not show whether or not glycerol has an effect on the control (no LPS) macrophages (M0s) in Figure EV3, which would be important to clarify if any effects from glycerol only occur in the LPS-treated condition.

RESPONSE:

We thank the reviewer for these helpful comments.

1. As suggested, all Seahorse data have now been normalized to cell number. The revised figures present oxygen consumption, proton efflux, and ATP production as values per 10⁵ cells.
2. We examined the effect of glycerol on control (M0) macrophages. High concentration of glycerol did not affect inflammatory gene expressions in M0 macrophages. In contrast, high extracellular glycerol reduced glycolytic ATP production in M0 macrophages, indicating that the metabolic effects of glycerol differ between M0 and M1 states. We have added these new figures to Fig. EV2I, EV3B, and EV3D.

(3) Figure 3 nicely shows regulation of relevant ETC genes by glycerol, and the specific regulation of complex III by glycerol and its inhibitors. The authors should demonstrate whether this regulation is specific to LPS treated macrophages and whether it occurs in control, M0 untreated macrophages. Furthermore, the result is striking because the G-3-P shuttle bypasses complex I and feeds electrons directly into the quinone pool. The authors should comment on how the specific regulation of complex III lines up well with the enzymes involved in G-3-P oxidation. Moreover, it would be logical since LPS activation causes nitric oxide-linked inhibition of complex I, thereby increasing reliance on glycerol oxidation upon LPS since the G-3-P shuttle feeds electrons straight into complex III via G3PDH.

RESPONSE:

We thank the reviewer for these insightful comments. As suggested, we examined whether glycerol-induced regulation of ETC genes occurs in untreated M0 macrophages. Consistent with the reviewer's expectation, glycerol did not increase the expression of ETC-related genes in M0 cells (new data added to Figure EV3B), indicating that this transcriptional upregulation is specific to LPS-activated inflammatory macrophages. Regarding the mechanistic point, we agree that the glycerol-3-phosphate shuttle donates electrons from GPD2 directly to the ubiquinone pool, thereby bypassing Complex I and relying predominantly on Complex III for electron flow. This is particularly relevant in LPS-activated macrophages, where nitric oxide-mediated inhibition of Complex I limits NADH-driven respiration and increases the relative contribution of glycerol oxidation. In line with this model, glycerol robustly upregulated Complex III-related genes in LPS-primed macrophages, and this response was blocked by GPD2 inhibition. We have added a paragraph in the Discussion (Page 14, line 19-).

(4) One note about the manuscript is that it is sometimes difficult to follow which time post-LPS treatment the authors are measuring and why. This is quite important given the temporal nature of the metabolic rewiring and histone acetylation (Ball et al. EMBO Rep (2025) 26: 982 - 1002; Lauterbach et al. Immunity 2019). In addition to clearly stating at what time each experiment was done and why, the authors should also examine the time dependence of the glycerol-induced increase of the pro-inflammatory response. Does this only happen at 24-48 hours, or what is the lag time between increased oxidation (earlier than 3 hours) and enhanced pro-inflammatory gene expression (I think measured at 24 hr) and cytokine release (48 hr.)

RESPONSE:

Thank you for this important comment. We now appreciate that in the original version we mixed data obtained from naïve M0 macrophages and LPS-primed inflammatory macrophages with different timing protocols, and we did not provide sufficient explanation of these experimental

conditions. We understand that this made it difficult for the reviewers to follow the temporal sequence and to interpret the effects of glycerol. In the revised manuscript, we have performed additional experiments, including ChIP assays and analyses of the inflammatory response in LPS-primed macrophages, and we have unified all main findings around our primary question of whether high concentrations of glycerol modulate pro-inflammatory macrophages. We now focus the central results on LPS-primed macrophages treated with glycerol, and for each relevant figure we have added schematic diagrams that clearly indicate the timing and conditions of LPS stimulation and glycerol treatment.

(5) The introduction is a bit outdated with regard to the field's understanding of mitochondrial energetics in macrophage polarization and may have some logical inconsistencies. If LPS shifts metabolism away from OxPhos, then why would increasing OxPhos enhance the pro-inflammatory response? A fuller picture including work from Ran et al [Nature Metabolism volume 5, pages804-820 (2023); which stands in contrast to the studies of Lauterbach], and work from Ball et al (cited in #4 and suggests that many metabolic phenotypes can accompany pro-inflammatory macrophage activation) would be helpful. The introduction should cite these works and give a more full picture of the field, and adjust the introduction to be more logically consistent with the findings.

RESPONSE:

We thank the reviewer for highlighting this important point. In the revised introduction, we incorporated recent findings to provide a more balanced and up-to-date overview of mitochondrial energetics in macrophage polarization. Specifically, we added references to Ran et al. (Nature Metabolism, 2023) and Ball et al. (EMBO Reports, 2025), which describe metabolic heterogeneity in LPS-activated macrophages, showing that mitochondrial respiration can be sustained or enhanced depending on context. This revision resolves the apparent inconsistency and aligns the Introduction with the current understanding of macrophage metabolism.

MINOR

(a) Given that the M+1 isotopologue for acetyl CoA is quite high, the authors should mention in the methods how they correct for natural heavy isotopes in the methods.

RESPONSE:

All isotopologue data were corrected for natural isotope abundance based on the known elemental composition of each metabolite. This information has now been added to the *Methods* section (Page 26, line 21-).

(b) The authors should cite their method for calculating ATP production rates from the XF

Analyzer. If using the company's kit, this is Desousa et al. *EMBO Reports* 2023 Oct 9;24(10):e56380.

RESPONSE:

The ATP production rate was calculated using the manufacturer's algorithm for the Seahorse XF Real-Time ATP Rate Assay. We have now cited the recommended reference (*Desousa et al., EMBO Reports* 2023 Oct 9; 24(10): e56380) in the *Methods* section.

Referee #3:

This manuscript is an investigation into the AQP3-mediated import of glycerol and its downstream metabolism in inflammatory macrophage metabolism and phenotype. It is a very interesting, timely and potentially physiologically relevant study. However, there are aspects of the study - in particular the non-physiologically high glycerol concentrations used in vitro, and that make the physiological relevance a little distant. Additionally, the metabolic results of the in vivo work are very exciting, somewhat unexpected, and not discussed in enough detail by the authors - given this unexpected nature (outlined below).

Major comments

1. Could the authors clarify what is meant in the Figure Legend for EV2 'Some cells were glucose-depleted overnight prior to the assay'. It isn't clear which assay, or whether this might produce some intra-assay variation that isn't otherwise obvious from the axis labels.

RESPONSE:

We thank the reviewer for this valuable comment. We agree that the description for this figure was insufficient and did not fully convey the intent of the experiments.

These assays were performed after incubating cells in glucose-depleted medium prior to the assay, followed by the addition of radiolabeled glucose or glucose analogs to measure uptake. In EV2C, we show that glycerol uptake into LPS-primed macrophages is not affected by the availability of extracellular glucose. For this, cells were maintained either in glucose-containing or glucose-depleted medium, and the uptake of radiolabeled glycerol was quantified.

To improve clarity, we have added a schematic representation of the experimental design for panels A–C and rearranged the order of the bars in panels B and C for consistency. The Figure Legend and corresponding text in the main manuscript have been revised accordingly to enhance readability and comprehension.

2. The authors state that on analysis, their FBS contains 1 mM glycerol. This is incredibly high and non-physiological for mammalian species - indeed in humans it is around 0.1 mM (Stinshoff et al. Clinical Chemistry 1977; Hagstrom-Toft et al. Endocrinology and Metabolism, 1997)). It is therefore confusing as to how this concentration was arrived at - whether bad batches of FBS were examined, or analytical error in measurement. Both would have implications for the manuscript as a whole. If the physiological concentration of glycerol of 0.1 mM is correct in the sera, then the relevance of this study is under very specific conditions (given the results in Figure 1A) - which would be likely observed in lipolytic adipose tissue.

RESPONSE:

We thank the reviewer for raising this important point regarding the glycerol concentration in FBS. In our original version, the glycerol concentration in three different lots of FBS was measured using the *Free Glycerol Reagent* kit (Sigma, F6428), and the average value was approximately 1 mM. In response to the reviewer’s comment, we re-evaluated the glycerol content using a more precise method, the *High Sensitivity Free Glycerol Assay Kit* (Sigma, MAK270). For this reassessment, we carefully quantified glycerol levels in three FBS lots stored in our laboratory. Unlike our previous measurement, where we prepared our own glycerol standard, this time we used the kit-provided glycerol standard to generate a standard curve and calculate concentrations. The measured glycerol levels were approximately 850 μ M in the FBS lots we are using for macrophage assay (HyClone). The raw data and lot details are provided below.

As for the rationale for using 1 mM glycerol in most experiments, this concentration was chosen based on our observation that serum glycerol levels in obese mice range between 300–600 μ M (as shown in the original Fig. 5C). Moreover, our preliminary experiments indicated that 300 μ M glycerol did not alter macrophage metabolic activity or inflammatory gene expression. Therefore, we hypothesized that glycerol influences pro-inflammatory macrophages only under conditions of elevated extracellular glycerol, such as in obesity. Hence, we performed our in vitro experiments mainly at 1 mM glycerol, a concentration slightly higher than the upper range of serum glycerol levels observed in our obese mice.

To address the reviewer’s concern, we have now added new data showing that lower concentrations of glycerol (≤ 300 μ M) did not affect inflammatory gene expression (Fig. EV3C) or ATP production (Fig. EV2H) in LPS-primed macrophages, and we describe this in the revised text. In addition, we now cite previous reports in the introduction indicating that glycerol concentrations in the local tissue microenvironment, particularly in lipolytic adipose tissue, can be considerably higher than those in plasma, providing a physiological context in which such elevated glycerol levels may be relevant.

glycerol assay (Sigma #MAK270)								
FBS	Lot #	Country	fluorescence intensity		mean	μ M	dilution	content (μ M)
HyClone AJ30699848	AJ30699848	USA	82188744	80084536	81136640	84.26	10	842.65
SIGMA 172012	22M284	Nicaragua	80770752	83445480	82108116	86.55	10	865.46
SIGMA 173012	BCCB5025	Ireland	67541600	66448064	66994832	54.68	10	546.84

3. In the initial experiment with a dose response for glycerol, the authors incubate the macrophages with the ¹⁴C-glycerol for 3 minutes. It isn't clear if the authors have data showing that this time is sufficient to reach steady-state concentrations in the cells, or whether if longer times were used, the profile of their experiment might change. A similar point is true for the ¹³C-glycerol experiment - the metabolites they report as being labelled - do they know that they have reached steadystate for these as well? It might well rely on the kinetics of glycerol kinase, and how readily the cells activate the glycerol for further metabolism in glycolysis

RESPONSE:

We thank the reviewer for this important comment. The 3-minute incubation used in the [¹⁴C]-glycerol uptake assay was not intended to reach steady-state, but rather to reflect the initial uptake rate of glycerol into macrophages. Our primary aim in the Fig. 1A experiment was to compare how macrophages take up glycerol at different extracellular concentrations, to estimate how changes in extracellular glycerol may influence intracellular levels, rather than to determine steady-state concentrations. This approach is consistent with the established methodology used in aquaporin (AQP)-mediated glycerol or water transport assays, where uptake is typically measured over a short time frame to capture the linear phase of channel-mediated diffusion (see e.g., Yang and Verkman, *J. Biol. Chem.* 1997; Ma et al., *J. Biol. Chem.* 2002). Thus, the short incubation time allows direct comparison of uptake kinetics between conditions without the confounding effects of downstream metabolism. We briefly explained this rationale in the Results section, citing previous studies.

4. When the authors examine the response of glycolysis and mitochondrial ATP generation mechanisms in response to LPS and then glycerol, they suggest that their results show that 'glycerol was able to restore mitochondrial ATP production...following LPS-induced suppression.' The suppression by LPS was recorded by the authors as around 3-fold reduction in OCR, accompanied by a 3-fold increase in EACR (EV3A). The change from this basal rate after glycerol was found to be ~33% increase in OCR (and mitoATP in 2A-C) and a further ~15% in EACR. The authors might consider altering their language to more accurately represent the increase in OCR by glycerol, which would need to be ~300% increase to restore mitochondrial ATP production.

RESPONSE:

We thank the reviewer for this helpful comment. We agree that our original wording overstated the effect of glycerol on mitochondrial ATP production. We have carefully revised the text to more accurately describe the results (Page 8, line 29-).

5. In the glycerol-starved control conditions, the authors were below their detection limit (although no data are shown to support this), which is 1 μ M or less. In Figure 2E-G they suggest that their results show that physiological glycerol levels are too low for inflammatory macrophages to utilise glycerol for energy production. However, for this experiment they did not test physiological levels of glycerol, but less than 1 μ M - 100-fold lower than the serum concentration. In order to make this conclusion, they need to test the effect of inhibitors on physiological levels - i.e. around 100 μ M.

RESPONSE:

Thank you for this constructive comment. Our original intention in Figure 2E–G was to demonstrate that high extracellular glycerol (1 mM) enhances OCR and mitochondrial ATP production in LPS-primed macrophages, and that this increase is suppressed by inhibition of the G3P shuttle. In contrast, under glycerol-starved conditions, inhibition of the G3P shuttle did not affect mitochondrial respiration, indicating that in the absence of exogenous glycerol this pathway contributes minimally to macrophage bioenergetics.

To address the reviewer's concern regarding physiological glycerol levels, we performed additional experiments using 100–300 μ M glycerol, which corresponds to the reported range of serum glycerol concentrations in control mice. In LPS-primed macrophages, these intermediate concentrations did not affect mitochondrial and glycolytic ATP production (new Fig. EV2H). Thus, our data indicates that glycerol can substantially alter mitochondrial respiration only at higher concentrations (1 mM).

We also agree with the reviewer that our previous wording that “at physiological glycerol levels (<100 μ M), inflammatory macrophages utilize minimal glycerol for energy production.” was

not appropriate and was not fully supported by the original dataset. We have therefore revised this statement in the Results sections to more accurately reflect our findings (page 9, line 18-). In addition, we re-quantified extracellular glycerol in the glycerol-starved condition using the High Sensitivity Free Glycerol Fluorometric Assay Kit (Sigma-Aldrich, MAK270). In this condition, glycerol levels were below the lowest standard (0.97 μ M). This information has been added to the Results section

6. In the GK experiment (EV2D-E) I could not find the extracellular concentrations of glycerol that were used in this assay - could this be corrected.

RESPONSE:

The extracellular glycerol concentrations used in the GK assay have now been specified in the Figure EV2E legend (under 1mM glycerol).

7. There is no section on analysis of the data arising from the metabolomic studies - this must be corrected

RESPONSE:

We have added a detailed description of the data analysis procedures for the metabolomic studies in the *Methods* section.

8. As a result of the successful inhibition of histone acetylation by the GPD2 inhibit, the authors point out that this suggests that the glycerol 3-phosphate shuttle is therefore implicated in this

process. It is unclear how this could mechanistically occur, given that it donates electrons directly to Ub. Could the authors suggest in the discussion how this might occur?

RESPONSE:

We thank the reviewer for this important mechanistic point. We agree that the mitochondrial glycerol 3-phosphate shuttle primarily donates electrons to ubiquinone, and therefore does not directly generate acetyl-CoA for histone acetylation. Specifically, inhibition of GPD2 may decrease mitochondrial NAD⁺ regeneration and thereby reduce cytosolic NADH/NAD⁺ balance, which is known to affect glycolytic and TCA cycle fluxes contributing to acetyl-CoA availability. In addition, reduced electron flow through the glycerol 3-phosphate shuttle could secondarily influence mitochondrial function and citrate export, which provides a major source of cytosolic acetyl-CoA. We have added this mechanistic consideration to the revised Discussion (Page 15, line 1-).

9. The authors cite two papers in support of elevated plasma glycerol levels in obese individuals - van der Merwe 1999 and 2001. The former describes glycerol in lean individuals, and the latter in lean, obese and diabetic women only. Not only that, the measurements taken show an increase in plasma glycerol from 55 μ M to 80 μ M between lean and obese individuals. This cannot be used to justify the hypothesis that the in vitro results using 1 mM glycerol could be recapitulated in obese animals. Could the authors profile glycerol in the plasma of the obese mice to show the contribution of the significant increase in lipolysis they show in EV4B to systemic concentrations?

RESPONSE:

We thank the reviewer for this insightful comment. We apologize for not making this information sufficiently clear in the original version. As noted, we have already measured plasma glycerol concentrations in both control and obese mice, and these data were presented in Figure 5C. The plasma glycerol concentration was approximately 150 μ M in control mice and ranged from 230 to 580 μ M in obese mice, indicating a clear elevation consistent with enhanced lipolysis (Fig. 5C). We have revised the Results section and figure legend to clarify this point.

Regarding the 1 mM glycerol used in our in vitro experiments, we agree that circulating plasma glycerol concentrations in humans are typically in the 50–100 μ M range, and this is now clearly stated in the text. However, microdialysis studies have demonstrated that interstitial glycerol concentrations in adipose tissue and skeletal muscle can reach millimolar levels, far exceeding those in plasma. For example, Maggs et al. (J Clin Invest, 1995) reported interstitial glycerol concentrations of approximately 0.63 mM in adipose tissue of lean individuals and up to 2.5 mM during fasting, which is 70–80 times higher than plasma levels (0.08 mM).

Given that macrophages residing in inflamed or lipolytic tissues are likely exposed to such

locally elevated glycerol levels, we used 1 mM glycerol in vitro to approximate these microenvironmental conditions. This rationale, together with supporting references, has been added in the revised introduction section.

10. The use of the AQP3 mouse is based around the inhibition of around 40% of glycerol uptake by DFP by macrophages in vitro and 50% in vivo in the AQP3 KO. Given this, it is surprising that the significant changes in fat content and metabolites were observed in the serum - indeed suggesting that loss of 50% of the glycerol transport function only in myeloid cells results in unexpectedly large metabolic defects - particularly decreased serum glycerol. This appears at odds with the hypothesis that macrophage import of glycerol is a key physiological feature, particularly with HFD, and this is not explained by the authors in the discussion. This is a critical discrepancy that should be resolved experimentally or explained in the discussion.

RESPONSE:

We thank the reviewer for this insightful comment. We agree that the ~40–50% reduction in macrophage glycerol uptake appears, at first sight, disproportionate to the marked changes in adiposity and serum metabolites. In the revised Discussion, we now clarify that we do not propose macrophages as the major quantitative sink for circulating glycerol. Instead, our data support a model in which AQP3-dependent glycerol uptake acts as an immunometabolic signal that amplifies inflammatory activation of WAT macrophages under HFD. Even a partial reduction in this pathway in AQP3 cKO mice markedly attenuated WAT inflammation and was associated with reduced weight gain. We also now explicitly interpret the lower serum glycerol levels in AQP3 cKO mice as secondary to reduced WAT inflammation and lipolysis, rather than as a direct consequence of increased glycerol clearance by macrophages. Thus, the systemic metabolic changes are most likely indirect effects of dampened adipose tissue inflammation. Finally, we acknowledge in the Discussion that we did not assess locomotor activity, whole-body energy expenditure, or brown/beige adipocyte activation, and we note that comprehensive metabolic phenotyping will be required in future studies to fully define the systemic impact of macrophage-specific AQP3 inhibition (Page 15, line 21).

Minor comments

11. Contrary to the first line of the introduction, the metabolic pathway of glycerol is indeed well-described. If the authors wanted to say something different - perhaps that any specific metabolic pathways used in macrophages from glycerol (which is indeed not yet documented) - they should say this more specifically.

RESPONSE:

We thank the reviewer for this constructive comment. We agree that the general metabolic pathways of glycerol are well characterized. We have revised the sentence in the *Result section* to clarify this point (Page 7, line 4-6).

12. Y-axis in EV3A should read pmol/min

RESPONSE:

We have corrected.

13. Figure 3F x axis has 'TG' instead of thio in the vehicle condition

RESPONSE:

We thank the reviewer for identifying this error. We have corrected.

Dear Dr. Hara-Chikuma,

Thank you for the submission of your revised manuscript to our editorial offices. I have already forwarded to you the reports I received from the three referees that I asked to re-evaluate your study. Please find them again below. As you know, referees #1 and #2 now support publication of the revised manuscript in EMBO reports. In contrast, referee #3 indicates several remaining concerns.

After going through your preliminary point-by-points response (further revision plan), and after cross-commenting with referees #1 and #2, I have decided to proceed with the submission. Please address the remaining concerns of referee #3 in a final revised manuscript and/or in a final detailed point-by-point response, as indicated in your revision plan.

Moreover, I have the editorial requests below. Please also provide a p-b-p-response regarding these with your final submission.

Editorial requests:

- Please provide a final title with not more than 100 characters.
- Please provide the final abstract written in present tense throughout.
- Please reduce the number of keywords to five and order the manuscript sections like this, using only these names: Title page - Abstract - Keywords - Introduction - Results - Discussion - Methods - Data availability section - Acknowledgements (please include here all the funding information) - Disclosure and Competing Interests Statement - References - Figure legends - Expanded View Figure legends

Thus, please include the funding information with the Acknowledgements and remove the section regarding 'Supplementary Material'.

- We now use CRediT to specify the contributions of each author in the journal submission system. CRediT replaces the author contribution section. Please use the free text box to provide more detailed descriptions and do NOT provide your final manuscript text file with an author contributions section. See also our guide to authors (section 'Author contributions'): <https://link.springer.com/journal/44319/submission-guidelines#cms-Revised-submissions>

- Please remove the list of abbreviations from the manuscript text file. Please define each abbreviation upon first mention in the manuscript text.

- Please check again that the number "n" for how many independent experiments were performed, their nature (biological versus technical replicates), the bars and error bars (e.g. SEM, SD) and the test used to calculate p-values is indicated in the respective figure legends (main, EV and Appendix figures). Please also check that all the p-values are explained in the legend, and that these fit to those shown in the figure. Please provide statistical testing where applicable. Please avoid the phrase 'independent experiment' but clearly state if these were biological or technical replicates. Please also indicate (e.g. with n.s.) if testing was performed, but the differences are not significant. In case n=2, please show the data as separate datapoints without error bars and statistics. See also:

<https://link.springer.com/journal/44319/submission-guidelines#cms-Figure-and-data-presentation>

- Please upload the Reagents and Tools Table as separate file and remove it from the main manuscript text. More information on how to adhere to this format as well as downloadable templates (.doc) for the Reagents and Tools Table can be found in our author guidelines (section 'Structured Methods'):

<https://link.springer.com/journal/44319/submission-guidelines#cms-Manuscript-organisation-and-formatting>

- Please remove now the referee access information from the Data Availability section and make sure that the datasets are public latest upon online publication of the manuscript. Please also add a direct link to the datasets. Please remove all other text regarding information availability from this section. Here only information regarding externally deposited datasets should be provided.

In addition, I would need from you uploaded separately:

- a short, two-sentence summary of the manuscript (not more than 35 words).
- two to four short (!) bullet points highlighting the key findings of your study (two lines each).
- a schematic summary figure as separate file that provides a sketch of the major findings (not a data image) in jpeg or tiff format

(with the exact width of 550 pixels and a height of not more than 400 pixels) that can be used as a visual synopsis on our website.

I look forward to seeing the further revised version of your manuscript when it is ready. Please let me know if you have questions regarding the revision.

Best,

Referee #1:

the authors have addressed my main concerns.

Referee #2:

The authors have appropriately addressed my concerns in their resubmitted manuscript.

Referee #3:

The revised manuscript from the authors has now shifted its focus to the role of glycerol on LPS-primed macrophage metabolism. Despite a significant amount of work to refocus the manuscript, questions remain that have not been fully resolved.

1. In particular, the comments regarding glycerol concentrations have not been sufficiently dealt with. The authors spend time within the introduction discussing the serum concentrations (see below for comment on accuracy again), which remains difficult to justify given that they also comment that 0.6 mM is high for serum, but they want to use 1 mM, without physiological justification in the serum. However, their justification lies in conditions observed in adipose tissues under lipolytic conditions. They need to be clear that they are modelling this environment.
2. In the introduction (first page), the authors have the statement that serum glycerol concentrations are significantly elevated during obesity or prolonged exercise, to values between 0.3mM and 1mM, and cite two sources for this. Again, these papers do not support this statement - the first (Robergs and Griffin, 1998) specifically says that serum glycerol concentrations can increase to 0.3mM during prolonged exercise or caloric restriction, providing the lower bound of the authors statement. However, the other paper (Jensen et al 2001) was performed on healthy, non-obese individuals, and was a study on fasting and reports glycerol turnover, rather than concentrations. Their statement is therefore not supported by the literature that they cite.
3. In their response to Referee #1, the authors rejected the use of dialyzed FBS due to unspecified difficulties in data interpretation. It is considered best practice to dialyze FBS when performing experiments in which metabolite concentrations are important. The authors' solution - to remove FBS entirely for 'at least 1 h before each assay' has the potential for significant additional complications due to the loss of growth factors, cytokines and chemokines, hormone-bound proteins and other factors such as lipoproteins.
4. In their response to Referees, the authors show their glycerol assay data. This level of transparency is appreciated, although it does also show issues with the assay itself. With the concentrations between 0-125 uM, the assay appears non-linear above 62.5 uM, therefore making measurements above this concentration inaccurate as it is beyond the linear response. This can be seen clearly in the second calibration curve shown, in which the lowest concentrations (0-7.8 uM) are assessed with a linear response. The concentration measured for two of the sera (measured at 84-86 uM after 1:10 dilution) are therefore likely still a significant overestimate.
5. The authors' response to Referee #3 point 8 makes no biochemical sense. With regards to inhibition of GPD2 decreasing mitochondrial NAD⁺ regeneration, reducing cytosolic NADH:NAD⁺ balance, which in turn affect glycolytic and TCA cycle fluxes - inhibition of GPD2 can indeed have expected effects on cytosolic and mitochondrial redox homeostasis - in the cytosol it could have effects on pyruvate to lactate equilibrium. More importantly perhaps, in the mitochondria the inhibition of GPD2 would reduce the electrons flow into ubiquinone, which thereby could allow for increased NADH oxidation by complex I which would increase (not decrease) mitochondrial NAD⁺ regeneration. The hypothesized effect of this if anything would be to increase acetyl-coA generation through PDH, not decrease it. Therefore, the question remains - given their hypothesis, could the authors please suggest in their discussion section how this might occur? Another potential means of the GPD2 inhibitor having this effect is through possible off-target cysteine reactivity (see Burger et al. Scientific Reports 2020) - given that the authors used a single inhibitor for GPD2, unexpected reactivities cannot be overlooked.

Referee #3

The revised manuscript from the authors has now shifted its focus to the role of glycerol on LPS-primed macrophage metabolism. Despite a significant amount of work to refocus the manuscript, questions remain that have not been fully resolved.

1. In particular, the comments regarding glycerol concentrations have not been sufficiently dealt with. The authors spend time within the introduction discussing the serum concentrations (see below for comment on accuracy again), which remains difficult to justify given that they also comment that 0.6 mM is high for serum, but they want to use 1 mM, without physiological justification in the serum. However, their justification lies in conditions observed in adipose tissues under lipolytic conditions. They need to be clear that they are modelling this environment.

Response:

We thank the referee for this important point. We agree that our revised Introduction did not sufficiently distinguish between circulating (serum) glycerol concentrations and the local glycerol levels expected in a lipolytic adipose-tissue microenvironment, which is the physiological context we model *in vitro*. We have therefore rewritten the Introduction to state explicitly that our *in vitro* experiments aim to model a glycerol-rich, lipolytic tissue niche and to determine how elevated extracellular glycerol shapes macrophage metabolism and pro-inflammatory programs (page 4, lines 23–25).

We have also added the statement of circulating glycerol: published mouse data report ~0.2–0.3 mM in lean controls and ~0.3–0.4 mM in obese animals (Iena et al., 2020). Accordingly, we clarified in the Results that 1 mM glycerol was chosen to approximate a glycerol-rich lipolytic tissue microenvironment that can exceed circulating levels, (page 7, lines 28-31).

2. In the introduction (first page), the authors have the statement that serum glycerol concentrations are significantly elevated during obesity or prolonged exercise, to values between 0.3mM and 1mM, and cite two sources for this. Again, these papers do not support this statement - the first (Robergs and Griffin, 1998) specifically says that serum glycerol concentrations can increase to 0.3mM during prolonged exercise or caloric restriction, providing the lower bound of the authors statement. However, the other paper (Jensen et al 2001) was performed on healthy, non-obese individuals, and was a study on fasting and reports glycerol turnover, rather than concentrations. Their statement is therefore not supported by the literature that they cite.

Response:

We thank the referee for pointing out that our previous statement and citations regarding circulating (serum/plasma) glycerol concentrations were not appropriately supported. We agree that the range “0.3–1 mM” was overstated based on the literature we cited. As the referee notes, Robergs and Griffin (1998) supports that circulating glycerol can rise to approximately ~0.3 mM under lipolytic conditions such as prolonged exercise or caloric restriction, whereas Jensen et al. (2001) focuses on glycerol turnover during fasting in healthy individuals and does not provide the concentration range needed to support our statement. We have therefore removed the unsupported upper range (1 mM), revised the Introduction to state that circulating glycerol can increase to ~0.3 mM in humans under lipolytic conditions, and retained Robergs and Griffin (1998) as the appropriate reference while removing Jensen et al. (2001) from this context (page 3, lines 8-10).

3. In their response to Referee #1, the authors rejected the use of dialyzed FBS due to unspecified difficulties in data interpretation. It is considered best practice to dialyze FBS when performing experiments in which metabolite concentrations are important. The authors' solution - to remove FBS entirely for 'at least 1 h before each assay' has the potential for significant additional complications due to the loss of growth factors, cytokines and chemokines, hormone-bound proteins and other factors such as lipoproteins.

Response:

We thank the referee for this methodological point. We agree that dialyzed FBS is often considered best practice when extracellular metabolite concentrations are a critical experimental variable, and we appreciate the concern that serum withdrawal may introduce confounding effects due to loss of serum-derived bioactive factors. In the revised manuscript, following Referee #1's request, we added control experiments to directly assess the impact of FBS dialysis and FBS removal in our macrophage system. LPS-primed macrophages (24 h) were further incubated for 24 h in (i) complete medium containing regular FBS, (ii) complete medium containing dialyzed FBS, or (iii) FBS-free medium. Under these conditions, expression of the inflammatory markers TNF- α and IL-1 β was comparable across media conditions (Fig. EV1C). Moreover, mitochondrial and glycolytic ATP production rates measured after 24 h incubation in FBS-free medium were similar to those in serum-containing conditions (Fig. EV2G). These data indicate that, within the experimental timeframe and for the primary readouts used in this study, substituting regular FBS with dialyzed FBS or FBS removal

does not measurably alter inflammatory gene expression or cellular bioenergetics in LPS-primed macrophages. Therefore, to define the direct impact of extracellular glycerol while minimizing background glycerol and other low-molecular-weight components present in FBS, we performed glycerol-addition and metabolic assays under FBS-free conditions.

Furthermore, we recognized that the results obtained using dialyzed FBS were not described with sufficient clarity in the previous version. We have now explicitly integrated these findings into the Results section (page 6, lines 13-16) to make the rationale for our experimental conditions more transparent.

4. In their response to Referees, the authors show their glycerol assay data. This level of transparency is appreciated, although it does also show issues with the assay itself. With the concentrations between 0-125 μM , the assay appears non-linear above 62.5 μM , therefore making measurements above this concentration inaccurate as it is beyond the linear response. This can be seen clearly in the second calibration curve shown, in which the lowest concentrations (0-7.8 μM) are assessed with a linear response. The concentration measured for two of the sera (measured at 84-86 μM after 1:10 dilution) are therefore likely still a significant overestimate.

Response:

We thank the referee for this careful assessment of our calibration curves and for pointing out that some of our earlier measurements were performed outside the linear response range of the assay. We agree that quantification should be restricted to the linear range.

We therefore repeated the glycerol measurements using higher dilutions of FBS (1:50, 1:100, and 1:200) and adjusted sample input such that fluorescence values fell within the linear portion of the standard curve. Importantly, the estimated glycerol concentration calculated from these serial dilutions was highly consistent, demonstrating dilution linearity. Across the dilutions that fell within the assay's linear range, we obtained a mean glycerol concentration of 832.9 μM for the FBS lot used in our macrophage assays. This value is in close agreement with the value reported in Fig. EV1B in the revised manuscript (842 μM), and therefore the figure does not require modification.

Standard curve							
fluorescence intensity		mean	μM				
7452066	7653508	7552787	0				
10400891	10703764	10552328	1.953125				
12747114	13511308	13129211	3.90625				
17974734	18416276	18195505	7.8125				
27716064	28586322	28151193	15.625				
43848356	46180384	45014370	31.25				
FBS	dilution	fluorescence intensity		mean	μM	x dilution	μM (final)
HyClone AJ30699848	1/50	28265974	28879890	28572932	16.891	50	844.5
	1/100	18538480	18476004	18507242	8.482	100	848.2
	1/200	13133264	13448478	13290871	4.124	200	824.9

5. The authors' response to Referee #3 point 8 makes no biochemical sense. With regards to inhibition of GPD2 decreasing mitochondrial NAD⁺ regeneration, reducing cytosolic NADH:NAD⁺ balance, which in turn affect glycolytic and TCA cycle fluxes - inhibition of GPD2 can indeed have expected effects on cytosolic and mitochondrial redox homeostasis - in the cytosol it could have affects on pyruvate to lactate equilibrium. More importantly perhaps, in the mitochondria the inhibition of GPD2 would reduce the electrons flow into ubiquinone, which thereby could allow for increased NADH oxidation by complex I which would increase (not decrease) mitochondrial NAD⁺ regeneration. The hypothesized effect of this if anything would be to increase acetyl-coA generation through PDH, not decrease it. Therefore, the question remains - given their hypothesis, could the authors please suggest in their discussion section how this might occur? Another potential means of the GPD2 inhibitor having this effect is through possible off-target cysteine reactivity (see Burger et al. Scientific Reports 2020) - given that the authors used a single inhibitor for GPD2, unexpected reactivities cannot be overlooked.

Response:

We thank the referee for this important mechanistic clarification. We agree that our previous explanation was imprecise and have removed the incorrect statement regarding NAD⁺ regeneration. We revised the Discussion to acknowledge that reduced electron input into the ubiquinone pool upon GPD2 inhibition could, in principle, permit

increased complex I-mediated NADH oxidation and support PDH flux (page 14, lines 2–). To explain why GPD2 inhibition nonetheless attenuates histone H3 acetylation in our system, we now discuss a model in which blocking GPD2 blunts glycerol-dependent respiratory/ATP output and downstream citrate export, thereby limiting nucleo-cytosolic acetyl-CoA availability for histone acetyltransferases; potential contributions from altered cytosolic redox balance (pyruvate–lactate equilibrium) are also noted.

Finally, as the referee pointed out, we now explicitly state the limitation that GPD2 inhibitor KM04416 is an isothiazolinone derivative with potential thiol/cysteine reactivity and therefore off-target effects cannot be excluded (Burger et al., *Sci Rep* 2020). Accordingly, we tempered our interpretation and noted that genetic validation and/or orthogonal tools will be required to establish pathway specificity.

Mariko Hara-Chikuma
School of Medicine, Keio University
Department of Pharmacology
35 Shinano-machi
35 shinanomachi
Shinjuku, TOKYO 160-8582
Japan

Dear Dr. Hara-Chikuma,

Thank you for the submission of your final manuscript to EMBO reports. I have now received the report from the referee that I asked to evaluate the revised study, which can be found at the end of this email. As you will see, the referee now fully supports publication of your manuscript.

As also all the editorial requests have been adequately addressed, I am very pleased to accept your manuscript for publication in the next available issue of EMBO reports. Thank you for your contribution to our journal.

You may qualify for financial assistance for your publication charges - either via a Springer Nature fully open access agreement or an EMBO initiative. Check your eligibility: <https://link.springer.com/journal/44319/how-to-publish-with-us>

Yours sincerely,

Referee #3:

The authors have now appropriately responded to all my concerns, and I feel the manuscript now has a narrative supported by the data.

>>> Please note that it is EMBO Reports policy for the transcript of the editorial process (containing referee reports and your response letter) to be published as an online supplement to each paper. If you do NOT want this, you will need to inform the Editorial Office via email immediately. More information is available here: <https://link.springer.com/partners/embo-press/editorial-policies#Peer%20review>